# ENERGY-GUIDED ENTROPIC NEURAL OPTIMAL TRANSPORT

**Petr Mokrov**[1]**, Alexander Korotin**[1,2]**, Alexander Kolesov**[1]**,
Nikita Gushchin**[1]**, Evgeny Burnaev**[1,2]
[1]Skolkovo Institute of Science and Technology, *Moscow, Russia*
[2]Artificial Intelligence Research Institute, *Moscow, Russia*
`{petr.mokrov,a.korotin}@skoltech.ru`

## ABSTRACT

Energy-based models (EBMs) are known in the Machine Learning community for decades. Since the seminal works devoted to EBMs dating back to the noughties, there have been a lot of efficient methods which solve the generative modelling problem by means of energy potentials (unnormalized likelihood functions). In contrast, the realm of Optimal Transport (OT) and, in particular, neural OT solvers is much less explored and limited by few recent works (excluding WGAN-based approaches which utilize OT as a loss function and do not model OT maps themselves). In our work, we bridge the gap between EBMs and Entropy-regularized OT. We present a novel methodology which allows utilizing the recent developments and technical improvements of the former in order to enrich the latter. From the theoretical perspective, we prove generalization bounds for our technique. In practice, we validate its applicability in toy 2D and image domains. To showcase the scalability, we empower our method with a pre-trained StyleGAN and apply it to high-res AFHQ $512 \times 512$ unpaired I2I translation. For simplicity, we choose simple short- and long-run EBMs as a backbone of our Energy-guided Entropic OT approach, leaving the application of more sophisticated EBMs for future research. Our code is available at: `https://github.com/PetrMokrov/Energy-guided-Entropic-OT`

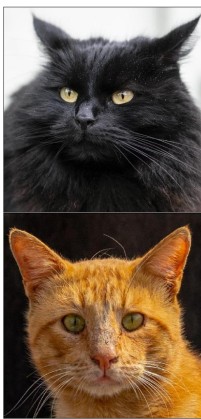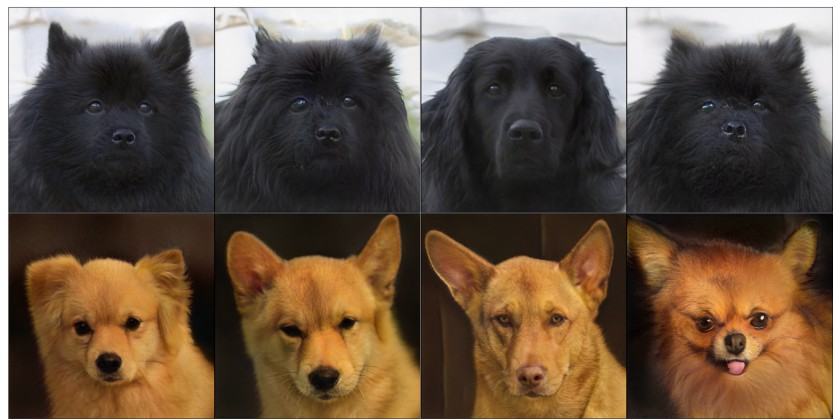

Figure 1: AFHQ $512 \times 512$ *Cat→Dog* unpaired translation by our Energy-guided EOT solver applied in the latent space of StyleGAN2-ADA. *Our approach does not need data2latent encoding. Left:* source samples; *right:* translated samples.

## 1 INTRODUCTION

The computational Optimal Transport (OT) field is an emergent and fruitful area in the Machine Learning research which finds its applications in generative modelling (Arjovsky et al., 2017; Gulrajani et al., 2017; Deshpande et al., 2018), domain adaptation (Nguyen et al., 2021; Shen et al., 2018; Wang et al., 2022), unpaired image-to-image translation (Xie et al., 2019; Hu et al.), datasets manipulation (Alvarez-Melis & Fusi, 2020), population dynamics (Ma et al., 2021; Wang et al., 2018), gradient flows modelling (Alvarez-Melis et al., 2022; Mokrov et al., 2021), barycenter estimation

(Korotin et al., 2022a; Fan et al., 2021). The majority of the applications listed above utilize OT as a loss function, e.g., have WGAN-like objectives which compare the generated (fake) and true data distributions. However, for some practical use cases, e.g., unpaired image-to-image translation (Korotin et al., 2023b), it is worth modelling the OT maps or plans by themselves.

The existing approaches which recover OT plans are based on various theoretically-advised techniques. Some of them (Makkuva et al., 2020; Korotin et al., 2021a) utilize the specific form of the cost function, e.g., squared Euclidean distance. The others (Xie et al., 2019; Lu et al., 2020) modify GAN objectives with additional OT regularizer, which results in biased OT solvers (Gazdieva et al., 2022, Thm. 1). The works (Fan et al., 2023; Korotin et al., 2023b; Rout et al., 2022) take advantage of dual OT problem formulation. They are capable of tackling *unbiased* large-scale continuous OT with general cost functions but may yield *fake* solutions (Korotin et al., 2023a). To overcome this issue, (Korotin et al., 2023a) propose to use strictly convex regularizers which guarantee the uniqueness of the recovered OT plans. And one popular choice which has been extensively studied both in discrete (Cuturi, 2013) and continuous (Genevay et al., 2016; Clason et al., 2021) settings is the **Entropy**. The well-studied methodological choices for modelling Entropy-regularized OT (EOT) include (a) stochastic dual maximization approach which prescribes alternating optimization of dual potentials (Seguy et al., 2018; Daniels et al., 2021) and (b) dynamic setup having connection to Schrödinger bridge problem (Bortoli et al., 2021; Gushchin et al., 2023; Chen et al., 2022). In contrast to the methods presented in the literature, we come up with an approach for solving EOT built upon EBMs.

**Contributions**. We propose a novel energy-based view on the EOT problem.

1. We take advantage of weak dual formulation for the EOT problem and distinguish the EBM-related nature of dual potential which originates due to this formulation (§3.1).

2. We propose theoretically-grounded yet easy-to-implement modifications to the standard EBMs training procedure which makes them capable of recovering the EOT plans (§3.2).

3. We establish generalization bounds for the EOT plans learned via our proposed method (§3.3).

4. We showcase our algorithm's performance on low- and moderate-dimensional toy setups and large-scale $512 \times 512$ images transfer tasks solved with help of a pre-trained StyleGAN (§5).

**Notations**. Throughout the paper, $\mathcal{X}$ and $\mathcal{Y}$ are compact subsets of the Euclidean space, i.e., $\mathcal{X} \subset \mathbb{R}^{D_x}$ and $\mathcal{Y} \subset \mathbb{R}^{D_y}$. The continuous functions on $\mathcal{X}$ are denoted as $\mathcal{C}(\mathcal{X})$. In turn, $\mathcal{P}(\mathcal{X})$ are the sets of Borel probability distributions on $\mathcal{X}$. Given distributions $\mathbb{P} \in \mathcal{P}(\mathcal{X})$ and $\mathbb{Q} \in \mathcal{P}(\mathcal{Y})$, $\Pi(\mathbb{P}, \mathbb{Q})$ designates the set of *couplings* between the distributions $\mathbb{P}$ and $\mathbb{Q}$, i.e., probability distributions on product space $\mathcal{X} \times \mathcal{Y}$ with the first and second marginals given by $\mathbb{P}$ and $\mathbb{Q}$, respectively. We use $\Pi(\mathbb{P})$ to denote the set of probability distributions on $\mathcal{X} \times \mathcal{Y}$ with the first marginal given by $\mathbb{P}$. The absolutely continuous probability distributions on $\mathcal{X}$ are $\mathcal{P}_{\text{ac}}(\mathcal{X})$. For $\mathbb{P} \in \mathcal{P}_{\text{ac}}(\mathcal{X})$ we use $\frac{d\mathbb{P}(x)}{dx}$ and $\frac{d\mathbb{Q}(y)}{dy}$ to denote the corresponding probability density functions. Given distributions $\mu$ and $\rho$ defined on a set $\mathcal{Z}$, $\mu \ll \rho$ means that $\mu$ is absolutely continuous with respect to $\rho$.

## 2 BACKGROUND

### 2.1 OPTIMAL TRANSPORT

The generic theory behind OT could be found in (Villani et al., 2009; Santambrogio, 2015). For the specific details on EOT, see (Genevay et al., 2016; Genevay, 2019).

Let $\mathbb{P} \in \mathcal{P}(\mathcal{X})$ and $\mathbb{Q} \in \mathcal{P}(\mathcal{Y})$. The primal OT problem due to Kantorovich (Villani et al., 2009) is:

$$\text{OT}_c(\mathbb{P}, \mathbb{Q}) \stackrel{\text{def}}{=} \inf_{\pi \in \Pi(\mathbb{P}, \mathbb{Q})} \int_{\mathcal{X} \times \mathcal{Y}} c(x, y) d\pi(x, y). \tag{1}$$

In the equation above, $c : \mathcal{X} \times \mathcal{Y} \to \mathbb{R}$ is a continuous *cost* function which reflects a practitioner's knowledge of how data from the source and target distribution should be aligned. Typically, the cost function $c(x, y)$ is chosen to be Euclidean norm $\|x - y\|_2$ yielding the 1-Wasserstain distance ($\mathbb{W}_1$) or halved squared Euclidean norm $\frac{1}{2}\|x - y\|_2^2$ yielding the square of 2-Wasserstein distance ($\mathbb{W}_2^2$).

The distributions $\pi^* \in \Pi(\mathbb{P}, \mathbb{Q})$ which minimize objective (1) are called the *Optimal Transport plans*. Problem (1) may have several OT plans (Peyré et al., 2019, Remark 2.3) and in order to impose the uniqueness and obtain a more tractable optimization problem, a common trick is to regularize (1) with strictly convex (w.r.t. distribution $\pi$) functionals $\mathcal{R} : \mathcal{P}(\mathcal{X} \times \mathcal{Y}) \to \mathbb{R}$.

**Entropy-regularized Optimal Transport**. In our work, we utilize the popular Entropic regularization (Cuturi, 2013) which has found its applications in various works (Solomon et al., 2015; Schiebinger et al., 2019; Rukhaia, 2021). This is mainly because of amenable sample complexity (Genevay, 2019, §3) and tractable dual representation of the Entropy-regularized OT problem which can be leveraged by, e.g., Sinkhorn's algorithm (Cuturi, 2013; Vargas et al., 2021). Besides, the EOT objective is known to be strictly convex (Genevay et al., 2016) thanks to the *strict* convexity of Entropy $H$ and KL divergence (Santambrogio, 2015; Nutz, 2021; Nishiyama, 2020) appearing in EOT formulations.

Let $\varepsilon > 0$. The EOT problem can be formulated in the following ways:

$$
\begin{cases}
\text{EOT}^{(1)}_{c,\varepsilon}(\mathbb{P},\mathbb{Q}) \\
\text{EOT}^{(2)}_{c,\varepsilon}(\mathbb{P},\mathbb{Q}) \\
\text{EOT}_{c,\varepsilon}(\mathbb{P},\mathbb{Q})
\end{cases}
\stackrel{\text{def}}{=} \min_{\pi \in \Pi(\mathbb{P},\mathbb{Q})} \int_{\mathcal{X}\times\mathcal{Y}} c(x,y)\mathrm{d}\pi(x,y) +
\begin{cases}
+\varepsilon\text{KL}\left(\pi\|\mathbb{P}\times\mathbb{Q}\right), & (2) \\
-\varepsilon H(\pi), & (3) \\
-\varepsilon\int_{\mathcal{X}} H(\pi(\cdot|x))\mathrm{d}\mathbb{P}(x). & (4)
\end{cases}
$$

These formulations are equivalent when $\mathbb{P}$ and $\mathbb{Q}$ are absolutely continuous w.r.t. the corresponding standard Lebesgue measures since $\text{KL}\left(\pi\|\mathbb{P}\times\mathbb{Q}\right) = -\int_{\mathcal{X}} H(\pi(\cdot|x))\mathrm{d}\mathbb{P}(x) + H(\mathbb{Q}) = -H(\pi) + H(\mathbb{Q}) + H(\mathbb{P})$. In other words, the equations (2), (3) and (4) are the same up to additive constants.

In the remaining paper, we will primarily work with the EOT formulation (4), and, henceforth, we will additionally assume $\mathbb{P} \in \mathcal{P}_{\text{ac}}(\mathcal{X})$, $\mathbb{Q} \in \mathcal{P}_{\text{ac}}(\mathcal{Y})$ when necessary.

Let $\pi^* \in \Pi(\mathbb{P},\mathbb{Q})$ be the solution of EOT problem. The measure disintegration theorem yields:
$$\mathrm{d}\pi^*(x,y) = \mathrm{d}\pi^*(y|x)\mathrm{d}\pi^*(x) = \mathrm{d}\pi^*(y|x)\mathrm{d}\mathbb{P}(x).$$
Distributions $\pi^*(\cdot|x)$ will play an important role in our analysis. In fact, they constitute the only ingredient needed to (stochastically) transform a source point $x \in \mathcal{X}$ to target samples $y_1, y_2, \cdots \in \mathcal{Y}$ w.r.t. EOT plan. We say that distributions $\{\pi^*(\cdot|x)\}_{x\in\mathcal{X}}$ are the *optimal conditional plans*.

**EOT problem as a weak OT (WOT) problem**. EOT problem (4) can be understood as the so-called *weak* OT problem (Gozlan et al., 2017; Backhoff-Veraguas et al., 2019). Given a *weak* transport cost $C : \mathcal{X} \times \mathcal{P}(\mathcal{Y}) \to \mathbb{R}$ which penalizes the displacement of a point $x \in \mathcal{X}$ into a distribution $\pi(\cdot|x) \in \mathcal{P}(\mathcal{Y})$, the weak OT problem is given by

$$\text{WOT}_C(\mathbb{P},\mathbb{Q}) \stackrel{\text{def}}{=} \inf_{\pi \in \Pi(\mathbb{P},\mathbb{Q})} \int_{\mathcal{X}} C(x,\pi(\cdot|x)) \underbrace{\mathrm{d}\pi(x)}_{=\mathrm{d}\mathbb{P}(x)}. \tag{5}$$

EOT formulation (4) is a particular case of weak OT problem (5) for weak transport cost:

$$C_{\text{EOT}}(x,\pi(\cdot|x)) = \int_{\mathcal{Y}} c(x,y)\mathrm{d}\pi(y|x) - \varepsilon H(\pi(\cdot|x)). \tag{6}$$

Note that if weak cost $C$ is *strictly* convex and lower semicontinuous, as it is the case for $C_{\text{EOT}}$, the solution for (5) exists and unique (Backhoff-Veraguas et al., 2019).

**Weak OT dual formulation of the EOT problem**. Similar to the case of classical Kantorovich OT (1), the weak OT problem permits the dual representation. Let $f \in \mathcal{C}(\mathcal{Y})$. Following (Backhoff-Veraguas et al., 2019, Eq. (1.3)) one introduces *weak $C$-transform* $f^C : \mathcal{X} \to \mathbb{R}$ by

$$f^C(x) \stackrel{\text{def}}{=} \inf_{\mu \in \mathcal{P}(\mathcal{Y})} \left\{ C(x,\mu) - \int_{\mathcal{Y}} f(y)\mathrm{d}\mu(y) \right\}. \tag{7}$$

For our particular case of EOT-advised weak OT cost (6), equation (7) reads as

$$f^{C_{\text{EOT}}}(x) = \min_{\mu \in \mathcal{P}(\mathcal{Y})} \left\{ \int_{\mathcal{Y}} c(x,y)\mathrm{d}\mu(y) - \varepsilon H(\mu) - \int_{\mathcal{Y}} f(y)\mathrm{d}\mu(y) \right\} \stackrel{\text{def}}{=} \min_{\mu \in \mathcal{P}(\mathcal{Y})} \mathcal{G}_{x,f}(\mu). \tag{8}$$

Note that the existence and uniqueness of the minimizer for (8) follows from Weierstrass theorem (Santambrogio, 2015, Box 1.1.) along with lower semicontinuity and strict convexity of $\mathcal{G}_{x,f}$ in $\mu$. The dual weak functional $F_C^w : \mathcal{C}(\mathcal{Y}) \to \mathbb{R}$ for primal WOT problem (5) is

$$F_C^w(f) \stackrel{\text{def}}{=} \int_{\mathcal{X}} f^C(x)\mathrm{d}\mathbb{P}(x) + \int_{\mathcal{Y}} f(y)\mathrm{d}\mathbb{Q}(y).$$

Thanks to the compactness of $\mathcal{X}$ and $\mathcal{Y}$, there is the strong duality (Gozlan et al., 2017, Thm. 9.5):

$$\text{EOT}_{c,\varepsilon}(\mathbb{P},\mathbb{Q}) = \sup_{f \in \mathcal{C}(\mathcal{Y})} \left\{ \int_{\mathcal{X}} \min_{\mu_x \in \mathcal{P}(\mathcal{Y})} \mathcal{G}_{x,f}(\mu_x)\mathrm{d}\mathbb{P}(x) + \int_{\mathcal{Y}} f(y)\mathrm{d}\mathbb{Q}(y) \right\} = \sup_{f \in \mathcal{C}(\mathcal{Y})} F_{C_{\text{EOT}}}^w(f). \tag{9}$$

We say that (9) is the *weak dual objective*. It will play an important role in our further analysis.

## 2.2 ENERGY-BASED MODELS

The EBMs are a fundamental class of deep Generative Modelling techniques (LeCun et al., 2006; Salakhutdinov et al., 2007) which parameterize distributions of interest $\mu \in \mathcal{P}(\mathcal{Y})$ by means of the Gibbs-Boltzmann distribution density:

$$\frac{\mathrm{d}\mu(y)}{\mathrm{d}y} = \frac{1}{Z} \exp\left(-E(y)\right). \tag{10}$$

In the equation above $E : \mathcal{Y} \to \mathbb{R}$ is the *Energy function* (negative unnormalized log-likelihood), and $Z = \int_{\mathcal{Y}} \exp(-E(y))\mathrm{d}y$ is the normalization constant, known as the partition function.

Let $\mu \in \mathcal{P}(\mathcal{Y})$ be a true data distribution which is accessible by samples and $\mu_\theta(y), \theta \in \Theta$ be a parametric family of distributions approximated using, e.g., a deep Neural Network $E_\theta$, which imitates the Energy function in (10). In EBMs framework, one tries to bring the parametric distribution $\mu_\theta$ to the reference one $\mu$ by optimizing the KL divergence between them. The minimization of KL $(\mu\|\mu_\theta)$ is done via gradient descent by utilizing the well-known gradient (Xie et al., 2016):

$$\frac{\partial}{\partial\theta}\mathrm{KL}\left(\mu\|\mu_\theta\right) = \int_{\mathcal{Y}} \frac{\partial}{\partial\theta}E_\theta(y)\mathrm{d}\mu(y) - \int_{\mathcal{Y}} \left[\frac{\partial}{\partial\theta}E_\theta(y)\right]\mathrm{d}\mu_\theta(y). \tag{11}$$

The expectations on the right-hand side of (11) are estimated by Monte-Carlo, which requires samples from $\mu$ and $\mu_\theta$. While the former are given, the latter are usually obtained via Unadjusted Langevin Algorithm (ULA) (Roberts & Tweedie, 1996). It iterates the discretized Langevin dynamics

$$Y_{l+1} = Y_l - \frac{\eta}{2}\frac{\partial}{\partial y}E_\theta(Y_l) + \sqrt{\eta}\xi_l\,, \quad \xi_l \sim \mathcal{N}(0,1), \tag{12}$$

starting from a simple prior distribution $Y_0 \sim \mu_0$, for $L$ steps, with a small discretization step $\eta > 0$. In practice, there have been developed a lot of methods, which improve or circumvent the procedure above by informative initialization (Hinton, 2002; Du & Mordatch, 2019), more sophisticated MCMC approaches (Lawson et al., 2019; Qiu et al., 2020; Nijkamp et al., 2022), regularizations (Du et al., 2021; Kumar et al., 2019), explicit auxiliary generators (Xie et al., 2018; Yin et al., 2022; Han et al., 2019; Gao et al., 2020). The application of these EBM improvements for the EOT problem is a fruitful avenue for future work. For a more in-depth discussion of the methods for training EBMs, see a recent survey (Song & Kingma, 2021).

## 3 TAKING UP EOT PROBLEM WITH EBMS

In this section, we connect EBMs and the EOT problem and exhibit our proposed methodology. At first, we present some theoretical results which characterize weak dual objective (9) and its optimizers (§3.1). Secondly, we develop the optimization procedure (§3.2) and corresponding algorithm capable of implicitly recovering EOT plans. Thirdly, we establish generalization bounds for our proposed method (§3.3). All proofs are situated in Appendix B.

### 3.1 ENERGY-GUIDED REFORMULATION OF WEAK DUAL EOT

We start our analysis by taking a close look at objective (9). The following proposition characterizes the inner $\min_{\mu_x}$ optimization problem arising in (9).

**Theorem 1** (Optimizer of weak $C_{\mathrm{EOT}}$-transform). *Let $f \in \mathcal{C}(\mathcal{Y})$ and $x \in \mathcal{X}$. Then inner weak dual objective $\min_{\mu \in \mathcal{P}(\mathcal{Y})} \mathcal{G}_{x,f}(\mu)$ (8) permits the unique minimizer $\mu_x^f$ which is given by*

$$\frac{\mathrm{d}\mu_x^f(y)}{\mathrm{d}y} \stackrel{def}{=} \frac{1}{Z(f,x)} \exp\left(\frac{f(y) - c(x,y)}{\varepsilon}\right), \tag{13}$$

*where $Z(f,x) \stackrel{def}{=} \int_{\mathcal{Y}} \exp\left(\frac{f(y)-c(x,y)}{\varepsilon}\right) \mathrm{d}y$.*

By substituting minimizer (13) to (8), we obtain the close form for the weak $C_{\mathrm{EOT}}$-transform:

$$f^{C_{\mathrm{EOT}}}(x) = \mathcal{G}_{x,f}(\mu_x^f) = -\varepsilon\log Z(f,x) = -\varepsilon\log\left(\int_{\mathcal{Y}} \exp\left(\frac{f(y) - c(x,y)}{\varepsilon}\right)\mathrm{d}y\right). \tag{14}$$

The equation (14) resembles $(c, \varepsilon)$-transform (Genevay, 2019, Eq. 4.15) appearing in standard *semi-dual EOT* formulation (Genevay, 2019, §4.3). For completeness, we shortly introduce

the dual EOT and semi-dual EOT problems in Appendix A, relegating readers to (Genevay, 2019) for a more thorough introduction. In short, it is the particular form of weak dual EOT objective, which **differs** from semi-dual EOT objective, and allows us to utilize EBMs, as we show in §3.2. Thanks to (14), objective (9) permits the reformulation:

$$\text{EOT}_{c,\varepsilon}(\mathbb{P}, \mathbb{Q}) = \sup_{f \in \mathcal{C}(\mathcal{Y})} F_{C_{\text{EOT}}}^w(f) = \sup_{f \in \mathcal{C}(\mathcal{Y})} \left\{ -\varepsilon \int_{\mathcal{X}} \log Z(f, x) \mathrm{d}\mathbb{P}(x) + \int_{\mathcal{Y}} f(y) \mathrm{d}\mathbb{Q}(y) \right\}. \quad (15)$$

For a given $f \in \mathcal{C}(\mathcal{Y})$, consider the distribution $\mathrm{d}\pi^f(x, y) \stackrel{\text{def}}{=} \mathrm{d}\mu_x^f(y) \mathrm{d}\mathbb{P}(x)$. We prove, that the optimization of weak dual objective (15) brings $\pi^f$ closer to the optimal plan $\pi^*$.

**Theorem 2** (Bound on the quality of the plan recovered from the dual variable). *For brevity, define the optimal value of (9) by $F_{C_{\text{EOT}}}^{w,*} \stackrel{\text{def}}{=} EOT_{c,\varepsilon}(\mathbb{P}, \mathbb{Q})$. For every $f \in \mathcal{C}(\mathcal{Y})$ it holds that*

$$F_{C_{\text{EOT}}}^{w,*} - F_{C_{\text{EOT}}}^w(f) = \varepsilon \int_{\mathcal{X}} KL\left(\pi^*(\cdot|x) \| \mu_x^f\right) d\mathbb{P}(x) = \varepsilon KL\left(\pi^* \| \pi^f\right). \quad (16)$$

From our Theorem 2 it follows that given an approximate maximizer $f$ of dual objective (15), one immediately obtains a distribution $\pi^f$ which is close to the optimal plan $\pi^*$. Actually, $\pi^f$ is formed by conditional distributions $\mu_x^f$ (Theorem 1), whose energy functions are given by $f$ (adjusted with transport cost $c$). Below we show that $f$ in (15) can be optimized analogously to EBMs as well.

### 3.2 Optimization procedure

Following the standard machine learning practices, we parameterize functions $f \in \mathcal{C}(\mathcal{Y})$ as neural networks $f_\theta$ with parameters $\theta \in \Theta$ and derive the loss function corresponding to (15) by:

$$L(\theta) \stackrel{\text{def}}{=} -\varepsilon \int_{\mathcal{X}} \log Z(f_\theta, x) \mathrm{d}\mathbb{P}(x) + \int_{\mathcal{Y}} f_\theta(y) \mathrm{d}\mathbb{Q}(y). \quad (17)$$

The conventional way to optimize loss functions such as (17) is the stochastic gradient ascent. In the following result, we derive the gradient of $L(\theta)$ w.r.t. $\theta$.

**Theorem 3** (Gradient of the weak dual loss $L(\theta)$). *It holds true that:*

$$\frac{\partial}{\partial \theta} L(\theta) = - \int_{\mathcal{X}} \int_{\mathcal{Y}} \left[ \frac{\partial}{\partial \theta} f_\theta(y) \right] \mathrm{d}\mu_x^{f_\theta}(y) \mathrm{d}\mathbb{P}(x) + \int_{\mathcal{Y}} \frac{\partial}{\partial \theta} f_\theta(y) \mathrm{d}\mathbb{Q}(y). \quad (18)$$

Formula (18) resembles the gradient of Energy-based loss, formula (11). This allows us to look at EOT problem (4) from the perspectives of EBMs. In order to emphasize the *novelty* of our approach, and, simultaneously, establish the deep connection between the optimization of weak dual objective in form (15) and EBMs, below we characterize the similarities and differences between standard EBMs optimization procedure and our proposed EOT-encouraged gradient ascent following $\partial L(\theta)/\partial \theta$.

**Differences**. In contrast to the case of EBMs, potential $f_\theta$, optimized by means of loss function $L$, does not represent an energy function by itself. However, the tandem of cost function $c$ and $f_\theta$ helps to recover the Energy functions of *conditional* distributions $\mu_x^{f_\theta}$:

$$E_{\mu_x^{f_\theta}}(y) = \frac{c(x, y) - f_\theta(y)}{\varepsilon}.$$

Therefore, one can sample from distributions $\mu_x^{f_\theta}$ following ULA (12) or using more advanced MCMC approaches (Girolami & Calderhead, 2011; Hoffman et al., 2014; Samsonov et al., 2022). In practice, when estimating (18), we need samples $(x_1, y_1), (x_2, y_2), \ldots (x_N, y_N)$ from distribution $\mathrm{d}\pi^{f_\theta}(x, y) \stackrel{\text{def}}{=} \mathrm{d}\mu_x^{f_\theta}(y) \mathrm{d}\mathbb{P}(x)$. They could be derived through the simple two-stage procedure:

1. Sample $x_1, \ldots x_N \sim \mathbb{P}$, i.e., derive random batch from the source dataset.
2. Sample $y_1|x_1 \sim \mu_{x_1}^{f_\theta}, \ldots, y_N|x_N \sim \mu_{x_N}^{f_\theta}$, e.g., performing Langevin steps (12).

**Similarities**. Besides a slightly more complicated two-stage procedure for sampling from generative distribution $\pi^{f_\theta}$, the gradient ascent optimization with (18) is similar to the gradient descent with

(11). This allows a practitioner to adopt the existing practically efficient architectures of EBMs, e.g., (Du & Mordatch, 2019; Du et al., 2021; Gao et al., 2021; Zhao et al., 2021), in order to solve EOT.

**Algorithm.** We summarize our findings and detail our optimization procedure in the Algorithm 1. The procedure is *basic*, i.e., for the sake of simplicity, we specifically remove all technical tricks which are typically used when optimizing EBMs (persistent replay buffers (Tieleman, 2008), temperature adjusting, etc.). Particular implementation details are given in the experiments section (§5).

We want to underline that our theoretical and practical setup allows performing theoretically-grounded **truly conditional** data generation by means of EBMs, which unlocks the data-to-data translation applications for the EBM community. Existing approaches leveraging such applications with Energy-inspired methodology lack theoretical interpretability, see discussions in §4.1.

---

**Algorithm 1:** Entropic Optimal Transport via Energy-Based Modelling

**Input** : Source and target distributions $\mathbb{P}$ and $\mathbb{Q}$, accessible by samples;
Entropy regularization coefficient $\varepsilon > 0$, cost function $c(x, y) : \mathbb{R}^{D_x} \times \mathbb{R}^{D_y} \to \mathbb{R}$;
number of Langevin steps $K > 0$, Langevin discretization step size $\eta > 0$;
basic noise std $\sigma_0 > 0$; potential network $f_\theta : \mathbb{R}^{D_y} \to \mathbb{R}$, batch size $N > 0$.

**Output** : trained potential network $f_{\theta^*}$ recovering optimal conditional EOT plans

**for** $i = 1, 2, \ldots$ **do**

  Derive batches $\{x_n\}_{n=1}^N = X \sim \mathbb{P}, \{y_n\}_{n=1}^N = Y \sim \mathbb{Q}$ of sizes N;

  Sample basic noise $Y^{(0)} \sim \mathcal{N}(0, \sigma_0)$ of size N;

  **for** $k = 1, 2, \ldots, K$ **do**

    Sample $Z^{(k)} = \{z_n^{(k)}\}_{n=1}^N$, where $z_n^{(k)} \sim \mathcal{N}(0, 1)$;

    Obtain $Y^{(k)} = \{y_n^{(k)}\}_{n=1}^N$ with Langevin step:

    $y_n^{(k)} \leftarrow y_n^{(k-1)} + \frac{\eta}{2\varepsilon} \cdot \texttt{stop\_grad}\left(\frac{\partial}{\partial y}\left[f_\theta(y) - c(x_n, y)\right]\big|_{y=y_n^{(k-1)}}\right) + \sqrt{\eta} z_n^{(k)}$

  $\widehat{L} \leftarrow -\frac{1}{N}\left[\sum_{y_n^{(K)} \in Y^{(K)}} f_\theta\left(y_n^{(K)}\right)\right] + \frac{1}{N}\left[\sum_{y_n \in Y} f_\theta(y_n)\right]$;

  Perform a gradient step over $\theta$ by using $\frac{\partial \widehat{L}}{\partial \theta}$;

---

### 3.3 GENERALIZATION BOUNDS FOR LEARNED ENTROPIC PLANS

Below, we sort out the question of how far a recovered plan is from the true optimal plan $\pi^*$.

In practice, the source and target distributions are given by empirical samples $X_N = \{x_n\}_{n=1}^N \sim \mathbb{P}$ and $Y_M = \{y_m\}_{m=1}^M \sim \mathbb{Q}$, i.e., finite datasets. Besides, the available potentials $f$ come from restricted functional class $\mathcal{F} \subset \mathcal{C}(\mathcal{Y})$, e.g., $f$ are neural networks. Therefore, in practice, we actually optimize the following *empirical counterpart* of the weak dual objective (15)

$$\max_{f \in \mathcal{F}} \widehat{F}_{C_{\text{EOT}}}^w(f) \stackrel{\text{def}}{=} \max_{f \in \mathcal{F}} \left\{ -\varepsilon \frac{1}{N} \sum_{n=1}^N \log Z(f, x_n) + \frac{1}{M} \sum_{m=1}^M f(y_m) \right\}.$$

and recover $\widehat{f} \stackrel{\text{def}}{=} \arg\max_{f \in \mathcal{F}} \widehat{F}_{C_{\text{EOT}}}^w(f)$. A question arises: **how close is $\pi^{\widehat{f}}$ to the OT plan $\pi^*$?**

Our Theorem 4 below characterizes the error between $\pi^{\widehat{f}}$ and $\pi^*$ arising due to *approximation* ($\mathcal{F}$ is restricted), and *estimation* (finite samples of $\mathbb{P}, \mathbb{Q}$ are available) errors. To bound the estimation error, we employ the well-known Rademacher complexity (Shalev-Shwartz & Ben-David, 2014, §26).

**Theorem 4** (Finite sample learning guarantees). *Denote the functional class of weak $C_{EOT}$-transforms corresponding to $\mathcal{F}$ by $\mathcal{F}^{C_{EOT}} = \{-\varepsilon \log Z(f, \cdot) : f \in \mathcal{F}\}$. Let $\mathcal{R}_N(\mathcal{F}, \mathbb{Q})$ and $\mathcal{R}_M(\mathcal{F}^{C_{EOT}}, \mathbb{P})$ denote the Rademacher complexities of functional classes $\mathcal{F}$ and $\mathcal{F}^{C_{EOT}}$ w.r.t. $\mathbb{Q}$ and $\mathbb{P}$ for sample sizes $N$ and $M$, respectively. Then the following upper bound on the error between $\pi^*$ and $\pi^{\widehat{f}}$ holds:*

$$\mathbb{E}\left[KL\left(\pi^* \| \pi^{\widehat{f}}\right)\right] \leq \overbrace{\varepsilon^{-1}\left[4\mathcal{R}_N(\mathcal{F}^{C_{EOT}}, \mathbb{P}) + 4\mathcal{R}_M(\mathcal{F}, \mathbb{Q})\right]}^{\text{Estimation error}} + \overbrace{\varepsilon^{-1}\left[F_{C_{EOT}}^{w,*} - \max_{f \in \mathcal{F}} F_{C_{EOT}}^w(f)\right]}^{\text{Approximation error}}. \quad (19)$$

*where the expectation is taken over random realizations of datasets $X_N \sim \mathbb{P}, Y_M \sim \mathbb{Q}$ of sizes $N, M$.*

We note that there exist many statistical bounds for EOT (Genevay, 2019; Genevay et al., 2019; Rigollet & Stromme, 2022; Mena & Niles-Weed, 2019; Luise et al., 2019; del Barrio et al., 2023), yet they are mostly about sample complexity of EOT, i.e., estimation of the OT **cost** value, or accuracy of the estimated barycentric projection $x \mapsto \int_{\mathcal{Y}} y \, d\pi^*(y|x)$ in the *non-parametric* setup. In contrast to these works, our result is about the estimation of the entire OT **plan** $\pi^*$ in the *parametric* setup. Our Theorem 4 could be improved by deriving explicit numerical bounds. This can be done by analyzing particular NNs architectures, similar to (Klusowski & Barron, 2018; Sreekumar et al., 2021). We leave the corresponding analysis to follow-up research.

## 4 RELATED WORKS

In this section, we look over existing works which are the most relevant to our proposed method. We divide our survey into two main parts. Firstly, we discuss the EBM approaches which tackle similar practical problem setups. Secondly, we perform an overview of solvers dealing with Entropy-regularized OT. The discussion of general-purpose OT solvers is available in Appendix A.2.

### 4.1 ENERGY-BASED MODELS FOR UNPAIRED DATA-TO-DATA TRANSLATION

Given a source and target domains $\mathcal{X}$ and $\mathcal{Y}$, accessible by samples, the problem of unpaired data-to-data translation (Zhu et al., 2017) is to transform a point $x \in \mathcal{X}$ from the source domain to corresponding points $y_1^x, y_2^x, \cdots \subset \mathcal{Y}$ from the target domain while "preserving" some notion of $x$'s content. In order to solve this problem, (Zhao & Chen, 2021; Zhao et al., 2021) propose to utilize a pretrained EBM of the target distribution $\mathbb{Q}$, initialized by source samples $x \sim \mathbb{P}$. In spite of plausibly-looking practical results, the theoretical properties of this approach remain unclear. Furthermore, being passed through MCMC, the obtained samples may lose the conditioning on the source samples. In contrast, our proposed approach is free from the aforementioned problems and can be tuned to reach the desired tradeoff between the conditioning power and data variability. The authors of (Xie et al., 2021) propose to cooperatively train CycleGAN and EBMs to solve unpaired I2I problems. However, in their framework, EBMs just help to stabilize the training of I2I maps and can not be considered as primal problem solvers.

### 4.2 ENTROPY-REGULARIZED OT

To the best of our knowledge, all continuous EOT solvers are based either on KL-guided formulation (2) (Genevay et al., 2016; Seguy et al., 2018; Daniels et al., 2021) or unconditional entropic one (3) with its connection to the Schrödinger bridge problem (Finlay et al., 2020; Bortoli et al., 2021; Gushchin et al., 2023; Chen et al., 2022; Shi et al., 2023). Our approach seems to be the first which takes advantage of conditional entropic formulation (4). Methods (Genevay et al., 2016; Seguy et al., 2018; Daniels et al., 2021) exploit dual form of (2), see (Genevay, 2019, Eq. 4.2), which is an unconstrained optimization problem w.r.t. a couple of dual potentials $(u, v)$. However, (Genevay et al., 2016; Seguy et al., 2018) do not provide a direct way for sampling from optimal conditional plans $\pi^*(y|x)$, since it requires the knowledge of target distribution $\mathbb{Q}$. In order to leverage this issue, (Daniels et al., 2021) proposes to employ a separate score-based model approximating $\mathbb{Q}$. At the inference stage (Daniels et al., 2021) utilizes MCMC sampling, which makes this work to be the closest to ours. The detailed comparison is given below:

1. The authors of (Daniels et al., 2021) optimize dual potentials $(u, v)$ following the dual form of (2). This procedure is unstable for small $\varepsilon$ as it requires the exponentiation of large numbers which are of order $\varepsilon^{-1}$. At the same time, a "small $\varepsilon$" regime is practically important for some downstream applications when one needs a close-to-deterministic plan between $\mathcal{X}$ and $\mathcal{Y}$ domains. On the contrary, our Energy-based approach does not require exponent computation and can be adapted for a small $\varepsilon$ by proper adjusting of ULA (12) parameters (step size, number of steps, etc.).

2. In (Daniels et al., 2021), it is mandatory to have *three* models, including a third-party score-based model. Our algorithm results in a *single* potential $f_\theta$ capturing all the information about the OT conditional plans and only optionally may be combined with an extra generative model (§5.3).

The alternative EOT solvers (Finlay et al., 2020; Bortoli et al., 2021; Gushchin et al., 2023; Chen et al., 2022; Vargas et al., 2021; Shi et al., 2023) are based on the connection between primal EOT (3) and the *Schrödinger bridge* problem. The majority of these works model the EOT plan as a

time-dependent stochastic process with learnable drift and diffusion terms, starting from $\mathbb{P}$ at the initial time and approaching $\mathbb{Q}$ at the final time. This requires resource-consuming techniques to solve stochastic differential equations. Moreover, the aforementioned methods work primarily with the quadratic cost and hardly could be accommodated for a more general case.

## 5 EXPERIMENTAL ILLUSTRATIONS

In what follows, we demonstrate the performance of our method on toy 2D scenario, Gaussian-to-Gaussian and high-dimensional AFHQ *Cat/Wild→Dog* image transformation problems solved using the latent space of a pretrained StyleGAN2-ADA (Karras et al., 2020). In the first two experiments the cost function is chosen to be squared halved $l_2$ norm: $c(x, y) = \frac{1}{2}\|x - y\|_2^2$, while in the latter case, it is more tricky and involves StyleGAN generator. An additional experiment with Colored MNIST images translation setup is considered in Appendix C.1.

Our code is written in `PyTorch` and publicly available at `https://github.com/PetrMokrov/Energy-guided-Entropic-OT`. The actual neural network architectures as well as practical training setups are disclosed in the corresponding subsections of Appendix D.

### 5.1 TOY 2D

We apply our method for 2D *Gaussian→Swissroll* case and demonstrate the qualitative results on Figure 2 for Entropy regularization coefficients $\varepsilon = 0.1, 0.001$. Figure 2b shows that our method succeeds in transforming source distribution $\mathbb{P}$ to target distribution $\mathbb{Q}$ for both Entropy regularization coefficients. In order to ensure that our approach learns optimal conditional plans $\pi^*(y|x)$ well, and correctly solves EOT problem, we provide Figures 2c and 2d. On these images, we pick several points $x \in \mathcal{X}$ and demonstrate samples from the conditional plans $\pi(\cdot|x)$, obtained either by our method $(\pi(\cdot|x) = \mu_x^{f_\theta})$ or by discrete EOT solver (Flamary et al., 2021). In contrast to our approach, the samples generated by the discrete EOT solver come solely from the training dataset. Yet these samples could be considered as a fine approximation of ground truth in 2D.

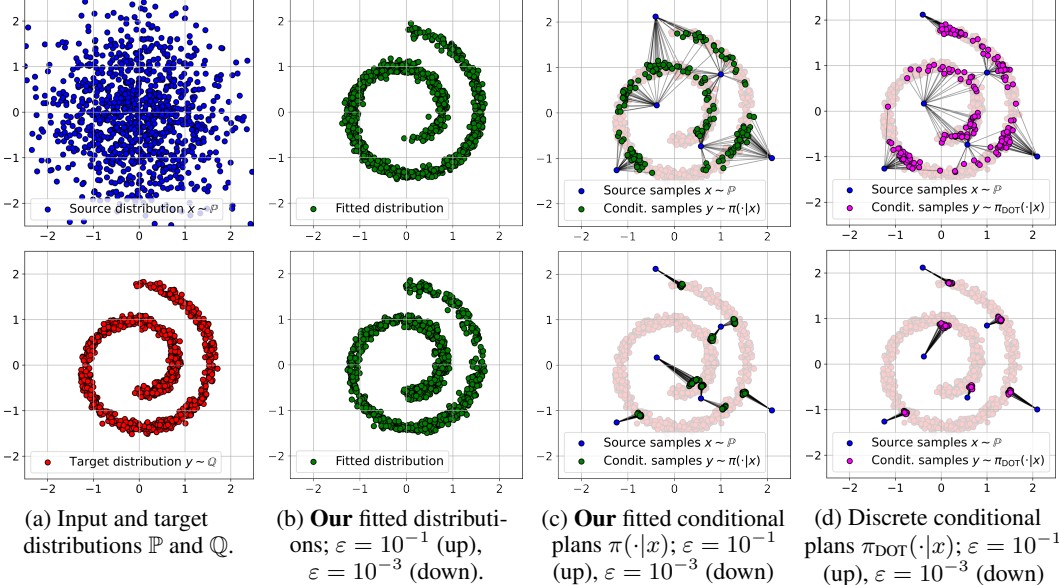

(a) Input and target distributions $\mathbb{P}$ and $\mathbb{Q}$.

(b) **Our** fitted distributions; $\varepsilon = 10^{-1}$ (up), $\varepsilon = 10^{-3}$ (down).

(c) **Our** fitted conditional plans $\pi(\cdot|x)$; $\varepsilon = 10^{-1}$ (up), $\varepsilon = 10^{-3}$ (down)

(d) Discrete conditional plans $\pi_{\text{DOT}}(\cdot|x)$; $\varepsilon = 10^{-1}$ (up), $\varepsilon = 10^{-3}$ (down)

Figure 2: Performance of Energy-guided EOT on *Gaussian→Swissroll* 2D setup.

### 5.2 GAUSSIAN-TO-GAUSSIAN

Here we validate our method in Gaussian-to-Gaussian transformation tasks in various dimensions ($D_x = D_y = 2, 16, 64, 128$), for which the exact optimal EOT plans are analytically known (Janati et al., 2020). We choose $\varepsilon = 0.1, 1, 10$ and compare the performance of our approach with those, described in §4.2. We report the $\text{B}\mathcal{W}_2^2$-UVP metric, see Appendix D.2 for the explanation, between the learned $\hat{\pi}$ and optimal $\pi^*$ plans in Table 1. As we can see, our method manages to recover the optimal plan rather well compared to baselines. Technical peculiarities are disclosed in Appendix D.

| Method | 2 | 16 | 64 | 128 | 2 | 16 | 64 | 128 | 2 | 16 | 64 | 128 |
|---|---|---|---|---|---|---|---|---|---|---|---|---|
| **Ours** | **0.01** | 0.2 | 0.56 | 0.97 | 0.03 | 0.1 | 0.26 | **0.31** | 0.11 | **0.09** | **0.18** | **0.29** |
| (Gushchin et al., 2023) | 0.02 | **0.08** | **0.19** | **0.34** | **0.006** | **0.04** | **0.12** | 0.33 | 0.22 | 0.15 | 0.4 | 0.86 |
| (Bortoli et al., 2021) | 1.97 | 4.22 | 3.44 | 5.87 | 0.88 | 1.29 | 2.32 | 2.43 | n/a | n/a | n/a | n/a |
| (Chen et al., 2022) *(Alt)* | 1.44 | 8.22 | 3.5 | 4.33 | 0.75 | 1.7 | 2.45 | 2.64 | n/a | n/a | n/a | n/a |
| (Chen et al., 2022) *(Joint)* | 0.45 | 4.8 | 5.6 | 5.28 | 0.07 | 0.21 | 0.34 | 0.58 | 0.88 | 2.12 | 2.77 | 3.45 |
| (Vargas et al., 2021) | 2.96 | 2.94 | 4.06 | 4.0 | 0.3 | 0.9 | 1.34 | 1.8 | **0.1** | 0.21 | 0.72 | 1.14 |
| (Daniels et al., 2021) | n/a | n/a | n/a | n/a | 0.92 | 1.36 | 4.62 | 5.33 | 3.22 | 5.57 | 3.13 | 4.98 |
| (Seguy et al., 2018) | 11.3 | 28.6 | 39.4 | 57.4 | 6.77 | 14.6 | 25.6 | 47.1 | 10.2 | 14.6 | 28.9 | 40.1 |
| | (a) $\varepsilon = 0.1$ | | | | (b) $\varepsilon = 1$ | | | | (c) $\varepsilon = 10$ | | | |

Table 1: Performance (B$\mathcal{W}_2^2$-UVP↓) of Energy-guided EOT (ours) and baselines on *Gaussian→Gaussian* tasks for dimensions $D = 2, 16, 64, 128$ and reg. coefficients $\varepsilon = 0.1, 1, 10$.

### 5.3 HIGH-DIMENSIONAL UNPAIRED IMAGE-TO-IMAGE TRANSLATION

In this subsection, we deal with the large-scale unpaired I2I setup. As with many other works in EBMs, e.g., (Zhao & Chen, 2021; Tiwary et al., 2022), we consider learning in the latent space. We pick a pre-trained StyleGANV2-Ada (Karras et al., 2020) generator $G$ for Dogs AFHQ $512 \times 512$ dataset and consider Cat→Dog (and Wild→Dog) unpaired translation. As $\mathbb{P}$, we use the dataset of images of cats (or wild); as $\mathbb{Q}$, we use $\mathcal{N}(0, I_{512})$, i.e., the latent distribution of the StyleGAN. We use our method with $\epsilon = 1$ to

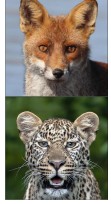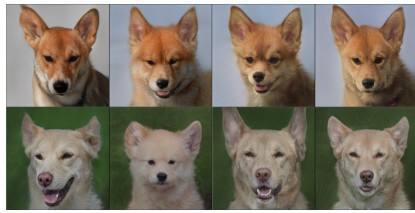

Figure 3: AFHQ $512 \times 512$ *Wild→Dog* unpaired I2I by our method in the latent space of StyleGAN2-ADA. *Left:* source; *right:* translated.

learn the EOT between $\mathbb{P}$ and $\mathbb{Q}$ with cost $\frac{1}{2}\|x - G(z)\|^2$, i.e., $\ell^2$ between the input image and the image generated from the latent code $z$. Note that our method trains **only one MLP** network $f_\theta$ acting on the latent space, which is then used for inference (combined with $G$). Moreover, our approach **does not need** a generative model of the source distribution $\mathbb{P}$, and **does not need** encoder (data2latent) networks. The qualitative results are provided in Figures 1 and 3. Our method allows us to translate the images from one domain to the other while maintaining the similarity with the input image. For more examples and qualitative comparisons, see Appendix C.2. For the quantitative analysis, we compare our approach with popular unpaired I2I models ILVR (Choi et al., 2021), SDEdit (Meng et al., 2022), EGSDE (Zhao et al., 2022), CycleGAN (Zhu et al., 2017), MUNIT (Huang et al., 2018) and StarGANv2 (Choi et al., 2020), the obtained FID metrics are reported in Table 2. As we can see, our approach achieves comparable-to-SOTA quality.

| Method | **Ours** | ILVR | SDEdit | EGSDE | CycleGAN | MUNIT | StarGAN v2 |
|---|---|---|---|---|---|---|---|
| Cat → Dog FID | 56.6 | 74.37 | 74.17 | 51.04 | 85.9 | 104.4 | 54.88 |
| Wild → Dog FID | 65.8 | 75.33 | 68.51 | 50.43 | - | - | - |

Table 2: Baselines FID[1] for Cat → Dog and Wild → Dog.

## 6 DISCUSSION

Our work paves a principled connection between EBMs and EOT. The latter is an emergent problem in generative modelling, with potential applications like unpaired data-to-data translation (Korotin et al., 2023b). Our proposed EBM-based learning method for EOT is theoretically grounded and we provide proof-of-concept experiments. We believe that our work will inspire future studies that will further empower EOT with recent EBMs capable of efficiently sorting out truly large-scale setups (Du et al., 2021; Gao et al., 2021; Zhao et al., 2021).

The **limitations** of our method roughly match those of basic EBMs. Namely, our method requires using MCMC methods for training and inference. This may be time-consuming. For the extended discussion of limitations, see Appendix F.

The **broader impact** of our work is the same as that of any generative modelling research. Generative models may be used for rendering, image editing, design, computer graphics, etc. and simplify the existing digital content creation pipelines. At the same time, it should be taken into account that the rapid development of generative models may also unexpectedly affect some jobs in the industry.

[1] FID scores of the baselines are taken from (Zhao et al., 2022). In order to estimate FID, we downscale the generated images to $256 \times 256$ for a fair comparison with the alternative methods.

## 7 ACKNOWLEDGEMENTS

The work was supported by the Analytical center under the RF Government (subsidy agreement 000000D730321P5Q0002, Grant No. 70-2021-00145 02.11.2021).

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

## A    EXTENDED BACKGROUND AND RELATED WORKS

### A.1    DUAL/SEMI-DUAL EOT PROBLEMS AND THEIR RELATION TO WEAK DUAL EOT PROBLEM

**Dual formulation of the EOT problem**. Primal EOT problem (2) has the dual reformulation (Genevay et al., 2016). Let $u \in \mathcal{C}(\mathcal{X})$ and $v \in \mathcal{C}(\mathcal{Y})$. Define the dual functional by

$$F_\varepsilon(u, v) \stackrel{\text{def}}{=} \int_\mathcal{X} u(x)\mathrm{d}\mathbb{P}(x) + \int_\mathcal{Y} v(y)\mathrm{d}\mathbb{Q}(y) - \varepsilon \int_{\mathcal{X} \times \mathcal{Y}} \exp\left(\frac{u(x) + v(y) - c(x, y)}{\varepsilon}\right) \mathrm{d}\left[\mathbb{P} \times \mathbb{Q}\right](x, y). \tag{20}$$

Then the strong duality holds (Genevay et al., 2016, Proposition 2.1), i.e.,

$$\mathrm{EOT}_{c,\varepsilon}^{(1)}(\mathbb{P}, \mathbb{Q}) = \sup_{u \in \mathcal{C}(\mathcal{X}), v \in \mathcal{C}(\mathcal{Y})} F_\varepsilon(u, v). \tag{21}$$

The $\sup$ here may not be attained in $\mathcal{C}(\mathcal{X}), \mathcal{C}(\mathcal{Y})$, yet it is common to relax the formulation and consider $u \in \mathcal{L}^\infty(\mathbb{P})$ and $v \in \mathcal{L}^\infty(\mathbb{Q})$ instead (Marino & Gerolin, 2020). In this case, the $\sup$ becomes $\max$ (Genevay, 2019, Theorem 7). The potentials $u^*, v^*$ which constitute a solution of the relaxed (21) are called the (Entropic) Kantorovich potentials. The optimal transport plan $\pi^*$ which solves the primal problem (2) could be recovered from a pair of Kantorovich potentials $(u^*, v^*)$ as

$$\mathrm{d}\pi^*(x, y) = \exp\left(\frac{u^*(x) + v^*(y) - c(x, y)}{\varepsilon}\right) \mathrm{d}\mathbb{P}(x)\mathrm{d}\mathbb{Q}(y). \tag{22}$$

From the practical viewpoint, the dual objective (21) is an unconstrained maximization problem which can be solved by conventional optimization procedures. The existing methods based on (21) as well as their limitations and drawbacks are discussed in the related works section, §4.2.

**Semi-dual formulation of the EOT problem**. Objective (21) is a convex optimization problem. By fixing a function $v \in \mathcal{C}(\mathcal{Y})$ and applying the first-order optimality conditions for the marginal optimization problem $\sup_{u \in \mathcal{C}(\mathcal{X})} F_\varepsilon(u, v)$, one can recover the solution of

$$v^{c,\varepsilon} \stackrel{\text{def}}{=} \arg\max_{u \in \mathcal{C}(\mathcal{X})} F_\varepsilon(u, v) \tag{23}$$

in the closed-form:

$$v^{c,\varepsilon}(x) = -\varepsilon \log\left(\int_\mathcal{Y} \exp\left(\frac{v(y) - c(x, y)}{\varepsilon}\right) \mathrm{d}\mathbb{Q}(y)\right). \tag{24}$$

Function $v^{c,\varepsilon}$ is called $(c, \varepsilon)$-*transform* (Genevay, 2019). By substituting the argument $u \in \mathcal{C}(\mathcal{X})$ of the $\max$ with $v^{c,\varepsilon}$ in equation (21) and performing several simplifications, one can recover the objective

$$\mathrm{EOT}_{c,\varepsilon}^{(1)}(\mathbb{P}, \mathbb{Q}) = \sup_{v \in \mathcal{C}(\mathcal{Y})} \int_\mathcal{X} v^{c,\varepsilon}(x)\mathrm{d}\mathbb{P}(x) + \int_\mathcal{Y} v(y)\mathrm{d}\mathbb{Q}(y). \tag{25}$$

This one is called the *semi-dual* formulation of EOT problem (Genevay, 2019, §4.3). It is not so popular as the classical dual problem (21) since the estimation of $v^{c,\varepsilon}$ is non-trivial, yet it has a direct relation to formulation (9), which forms the basis of our proposed method. In order to comprehend this relation, below we compare $(c, \varepsilon)$-transform and $\min_\mu \mathcal{G}_{x,f}(\mu)$, given by (14).

**Correspondence between (semi-) dual EOT and weak dual EOT**. As we already pointed out in the main part of the manuscript, equation (14) which then appears in weak dual objective (15) looks similar to $(c, \varepsilon)$-transform (24). The difference is the integration measure, which is $\mathbb{Q}$ in the case of (24) and the standard Lebesgue one in our case (14).

From the *theoretical* point of view, such dissimilarity is not significant. That is why semi-dual (25) and weak dual (9) optimization problems are expected to share such properties as convergence, existence of optimizers and so on. In particular, there is a relation between potentials $u, v$ that appear in (semi-) dual EOT problems (21, 25), and our optimized potential $f$ from (9).

Let $\mathbb{Q} \in \mathcal{P}_{\mathrm{ac}}(\mathcal{Y})$, and $E_\mathbb{Q} : \mathcal{Y} \to \mathbb{R}$ be the energy function of $\mathbb{Q}$, i.e., $\frac{\mathrm{d}\mathbb{Q}(y)}{\mathrm{d}y} \propto \exp\left(-E_\mathbb{Q}(y)\right)$. Consider the parameterization of potentials $v$ by means of $f$ as follows:

$$v(y) \leftarrow f(y) + \varepsilon E_\mathbb{Q}(y). \tag{26}$$

Then,

$$
\begin{aligned}
v^{c,\varepsilon}(x) &= -\varepsilon \log \left( \int_{\mathcal{Y}} \exp\left( \frac{v(y) - c(x,y)}{\varepsilon} \right) \mathrm{d}\mathbb{Q}(y) \right) \\
&= -\varepsilon \log \left( \int_{\mathcal{Y}} \exp\left( \frac{f(y) - c(x,y)}{\varepsilon} \right) \exp\left( E_{\mathbb{Q}}(y) \right) \frac{\mathrm{d}\mathbb{Q}(y)}{\mathrm{d}y} \mathrm{d}y \right) \\
&= -\varepsilon \log \left( \int_{\mathcal{Y}} \exp\left( \frac{f(y) - c(x,y)}{\varepsilon} \right) \exp\left( E_{\mathbb{Q}}(y) \right) \exp\left( -E_{\mathbb{Q}}(y) \right) \mathrm{d}y \right) + \mathrm{Const}(\mathbb{Q}) \\
&\stackrel{\text{see (14)}}{=} -\varepsilon \log Z(f,x) + \mathrm{Const}(\mathbb{Q}).
\end{aligned}
\tag{27}
$$

By substituting (26, 27) in (25) we obtain that semi-dual EOT objective (25) *recovers* our weak dual objective (15) up to reparameterization (26):

$$
\begin{aligned}
\mathrm{EOT}^{(1)}_{c,\varepsilon}(\mathbb{P}, \mathbb{Q}) &= \sup_{v \in \mathcal{C}(\mathcal{Y})} \left\{ \int_{\mathcal{X}} v^{c,\varepsilon}(x) \mathrm{d}\mathbb{P}(x) + \int_{\mathcal{Y}} v(y) \mathrm{d}\mathbb{Q}(y) \right\} \\
&= \underbrace{\mathrm{Const}_1(\mathbb{Q})}_{= \varepsilon H(\mathbb{Q})} + \sup_{f \in \mathcal{C}(\mathcal{Y}) - \varepsilon E_{\mathbb{Q}}} \left\{ \int_{\mathcal{X}} \left[ -\varepsilon \log Z(f,x) \right] \mathrm{d}\mathbb{P}(x) + \int_{\mathcal{Y}} f(y) \mathrm{d}\mathbb{Q}(y) \right\}.
\end{aligned}
$$

Strictly speaking, the optimization class $\mathcal{C}(\mathcal{Y}) - \varepsilon E_{\mathbb{Q}} = \{ f - \varepsilon E_{\mathbb{Q}}, f \in \mathcal{C}(\mathcal{Y}) \}$ in the equation above is different from $\mathcal{C}(\mathcal{Y})$ that appears in (15). However, under mild assumptions on $E_{\mathbb{Q}}$ the corresponding optimization problems are similar. One important consequence of the observed equivalence is the existence of optimal potential $f^*$ (not necessary to be continuous) which solves weak dual objective (9). It can be expressed through optimal Kantorovich potential $v^*$ by $f^* = v^* - \varepsilon E_{\mathbb{Q}}$.

From the *practical* point of view, the difference between (14) and (24) is much more influential. Actually, it is the particular form of internal weak dual problem solution (14) that allows us to utilize EBMs, see §3.2.

## A.2   DISCRETE AND CONTINUOUS OT SOLVERS REVIEW

**Discrete OT**. The discrete OT (DOT) is the specific domain in OT research area, which deals with distributions supported on finite discrete sets. There have been developed various methods for solving DOT problems (Peyré et al., 2019), the most efficient of which deals with discrete EOT (Cuturi, 2013). In spite of good theoretically-grounded convergence guarantees, it is hard to adopt the DOT solvers for out-of-distribution sampling and mapping, which limits their applicability in some real-world scenarios.

**Continuous OT**. In the continuous setting, the source and target distributions become accessible only by samples from (limited) budgets. In this case, OT plans are typically parameterized with neural networks and optimized with the help of SGD-like methods by deriving random batches from the datasets. The approaches dealing with such practical setup are called *continuous OT solvers*.

There exists a lot of continuous OT solvers (Makkuva et al., 2020; Korotin et al., 2021a; Fan et al., 2023; Xie et al., 2019; Rout et al., 2022; Gazdieva et al., 2022; Korotin et al., 2022b; Taghvaei & Jalali, 2019; Liu et al., 2023). However, the majority of these methods model OT as a deterministic map which for each input point $x$ assigns a single data point $y$ rather than distribution $\pi(y|x)$. Only a limited number of approaches are capable of solving OT problems which require stochastic mapping, and, therefore, potentially applicable for our EOT case. Here we exclude methods designed *specifically* for EOT and cover them in the further narrative.

The recent line of works (Korotin et al., 2023b;a; Asadulaev et al., 2024) considers dual formulation of weak OT problem (Gozlan et al., 2017) and comes up with $\max \min$ objective for various weak (Korotin et al., 2023b;a) and even general (Asadulaev et al., 2024) cost functionals. However, their proposed methodology requires the estimation of weak cost by samples, which complicates its application for EOT. An alternative concept (Xie et al., 2019) works with primal OT formulation (1) and lifts boundary source and target distribution constraints by WGAN losses. It also utilizes sample estimation of corresponding functionals and can not be directly adapted for EOT setup.

## B  PROOFS

### B.1  PROOF OF THEOREM 1

*Proof.* In what follows, we analyze the optimizers of the objective $\min_{\mu \in \mathcal{P}(\mathcal{Y})} \mathcal{G}_{x,f}(\mu)$ introduced in (8). Let $\mu \in \mathcal{P}(\mathcal{Y})$. We have:

$$
\begin{aligned}
\mathcal{G}_{x,f}(\mu) &= \int_{\mathcal{Y}} c(x,y)\mathrm{d}\mu(y) + \varepsilon \int_{\mathcal{Y}} \log \frac{\mathrm{d}\mu(y)}{\mathrm{d}y}\mathrm{d}\mu(y) - \int_{\mathcal{Y}} f(y)\mathrm{d}\mu(y) \\
&= \varepsilon \int_{\mathcal{Y}} \left( \frac{c(x,y) - f(y)}{\varepsilon} + \log \frac{\mathrm{d}\mu(y)}{\mathrm{d}y} \right) \mathrm{d}\mu(y) \\
&= \varepsilon \int_{\mathcal{Y}} \Big( \underbrace{-\log Z(f,x)}_{\text{does not depend on } y} + \underbrace{\log Z(f,x) - \frac{f(y)-c(x,y)}{\varepsilon}}_{= -\log \frac{\mathrm{d}\mu_x^f(y)}{\mathrm{d}y}} + \log \frac{\mathrm{d}\mu(y)}{\mathrm{d}y} \Big) \mathrm{d}\mu(y) \\
&= -\varepsilon \log Z(f,x) + \varepsilon \int_{\mathcal{Y}} \left( -\log \frac{\mathrm{d}\mu_x^f(y)}{\mathrm{d}y} + \log \frac{\mathrm{d}\mu(y)}{\mathrm{d}y} \right) \mathrm{d}\mu(y) \\
&= -\varepsilon \log Z(f,x) + \varepsilon \mathrm{KL}\left( \mu \| \mu_x^f \right).
\end{aligned} \tag{28}
$$

The last equality holds true thanks to the fact that $\mu_x^f \in \mathcal{P}_{\mathrm{ac}}(\mathcal{Y})$ and $\forall y \in \mathcal{Y} : \frac{\mathrm{d}\mu_x^f(y)}{\mathrm{d}y} > 0$. This leads to the conclusion that the absolute continuity of $\mu$ ($\mu \ll \lambda$, where $\lambda$ is the Lebesgue measure on $\mathcal{Y}$) is equivalent to the absolute continuity of $\mu$ w.r.t. $\mu_x^f$ ($\mu \ll \mu_x^f$). In particular, if $\mu \notin \mathcal{P}_{\mathrm{ac}}(\mathcal{Y})$, then the last equality in the derivations above reads as $+\infty = +\infty$. From (28) we conclude that $\mu_x^f = \arg\min_{\mu \in \mathcal{P}(\mathcal{Y})} \mathcal{G}_{x,f}(\mu)$. $\square$

### B.2  PROOF OF THEOREM 2

*Proof.* Recall that we denote the optimal value of weak dual objective (9) by $F_{C_{\mathrm{EOT}}}^{w,*}$. It equals to $\mathrm{EOT}_{c,\varepsilon}(\mathbb{P}, \mathbb{Q})$ given by formula (4), thanks to the strong duality, i.e.,

$$
F_{C_{\mathrm{EOT}}}^{w,*} = \int_{\mathcal{X} \times \mathcal{Y}} c(x,y)\mathrm{d}\pi^*(x,y) - \varepsilon \int_{\mathcal{X}} H(\pi^*(y|x))\mathrm{d}\mathbb{P}(x). \tag{29}
$$

For a potential $f \in \mathcal{C}(\mathcal{Y})$, our Theorem 1 yields:

$$
F_{C_{\mathrm{EOT}}}^w(f) = \int_{\mathcal{X}} \mathcal{G}_{x,f}(\mu_x^f)\mathrm{d}\mathbb{P}(x) + \int_{\mathcal{Y}} f(y)\mathrm{d}\mathbb{Q}(y) \stackrel{\mathrm{Eq.\ (14)}}{=} -\varepsilon \int_{\mathcal{X}} \log Z(f,x)\mathrm{d}\mathbb{P}(x) + \int_{\mathcal{Y}} f(y)\mathrm{d}\mathbb{Q}(y). \tag{30}
$$

In what follows, we combine (29) and (30):

$$
F_{C_{\mathrm{EOT}}}^{w,*} - F_{C_{\mathrm{EOT}}}^w(f) =
$$

$$
\int_{\mathcal{X} \times \mathcal{Y}} c(x,y)\mathrm{d}\pi^*(x,y) - \varepsilon \int_{\mathcal{X}} H(\pi^*(y|x))\mathrm{d}\mathbb{P}(x) + \varepsilon \int_{\mathcal{X}} \log Z(f,x) \underbrace{\mathrm{d}\mathbb{P}(x)}_{= \mathrm{d}\pi^*(x)} - \int_{\mathcal{Y}} f(y) \underbrace{\mathrm{d}\mathbb{Q}(y)}_{= \mathrm{d}\pi^*(y)} =
$$

$$
\int_{\mathcal{X} \times \mathcal{Y}} [c(x,y) - f(y)]\mathrm{d}\pi^*(x,y) - \varepsilon \int_{\mathcal{X}} H(\pi^*(y|x))\mathrm{d}\mathbb{P}(x) + \varepsilon \int_{\mathcal{X}} \log Z(f,x)\mathrm{d}\pi^*(x) =
$$

$$
-\varepsilon \int_{\mathcal{X} \times \mathcal{Y}} \underbrace{\frac{f(y) - c(x,y)}{\varepsilon}}_{= \log \exp\left( \frac{f(y)-c(x,y)}{\varepsilon} \right)} \mathrm{d}\pi^*(x,y) + \varepsilon \int_{\mathcal{X} \times \mathcal{Y}} \log \underbrace{Z(f,x)}_{= \int_{\mathcal{Y}} \exp\left( \frac{f(y)-c(x,y)}{\varepsilon} \right)\mathrm{d}y} \mathrm{d}\pi^*(x,y) - \varepsilon \int_{\mathcal{X}} H(\pi^*(y|x))\mathrm{d}\mathbb{P}(x) =
$$

$$
-\varepsilon \left\{ \int_{\mathcal{X} \times \mathcal{Y}} \log \Big[ \underbrace{\frac{1}{Z(f,x)} \exp \left( \frac{f(y) - c(x,y)}{\varepsilon} \right)}_{= \frac{\mathrm{d}\mu_x^f(y)}{\mathrm{d}y}} \Big] \mathrm{d}\pi^*(x,y) \right\} - \varepsilon \int_{\mathcal{X}} H(\pi^*(y|x))\mathrm{d}\mathbb{P}(x) =
$$

$$
-\varepsilon \int_{\mathcal{X} \times \mathcal{Y}} \log \left( \frac{\mathrm{d}\mu_x^f(y)}{\mathrm{d}y} \right) \mathrm{d}\pi^*(x,y) - \varepsilon \int_{\mathcal{X}} H(\pi^*(y|x))\mathrm{d}\mathbb{P}(x) =
$$

$$-\varepsilon \int_{\mathcal{X}} \int_{\mathcal{Y}} \log \left( \frac{\mathrm{d}\mu_x^f(y)}{\mathrm{d}y} \right) \mathrm{d}\pi^*(y|x) \underbrace{\mathrm{d}\pi^*(x)}_{=\mathrm{d}\mathbb{P}(x)} + \varepsilon \int_{\mathcal{X}} \int_{\mathcal{Y}} \log \left( \frac{\mathrm{d}\pi^*(y|x)}{\mathrm{d}y} \right) \mathrm{d}\pi^*(y|x)\mathrm{d}\mathbb{P}(x) =$$

$$\varepsilon \int_{\mathcal{X}} \left\{ \int_{\mathcal{Y}} \left[ \log \left( \frac{\mathrm{d}\pi^*(y|x)}{\mathrm{d}y} \right) - \log \left( \frac{\mathrm{d}\mu_x^f(y)}{\mathrm{d}y} \right) \right] \mathrm{d}\pi^*(y|x) \right\} \mathrm{d}\mathbb{P}(x) =$$

$$\varepsilon \int_{\mathcal{X}} \mathrm{KL} \left( \pi^*(\cdot|x) \| \mu_x^f \right) \mathrm{d}\mathbb{P}(x) .$$

In the last transition of the derivations above, we use the equality

$$\int_{\mathcal{Y}} \left[ \log \left( \frac{\mathrm{d}\pi^*(y|x)}{\mathrm{d}y} \right) - \log \left( \frac{\mathrm{d}\mu_x^f(y)}{\mathrm{d}y} \right) \right] \mathrm{d}\pi^*(y|x) = \mathrm{KL} \left( \pi^*(\cdot|x) \| \mu_x^f \right) .$$

It holds true due to the same reasons, as explained in the proof of Theorem 1, after formula (28).

We are left to show that $\int_{\mathcal{X}} \mathrm{KL} \left( \pi^*(\cdot|x) \| \mu_x^f \right) \mathrm{d}\mathbb{P}(x) = \mathrm{KL} \left( \pi^* \| \pi^f \right)$. Since $\mathrm{d}\pi^*(x,y) = \mathrm{d}\pi^*(y|x)\mathrm{d}\mathbb{P}(x)$ and $\mathrm{d}\pi^f(x,y) = \mathrm{d}\mu_x^f(y)\mathrm{d}\mathbb{P}(x)$, we derive:

$$\int_{\mathcal{X}} \mathrm{KL} \left( \pi^*(\cdot|x) \| \mu_x^f \right) \mathrm{d}\mathbb{P}(x) = \int_{\mathcal{X}} \int_{\mathcal{Y}} \log \left( \frac{\mathrm{d}\pi^*(y|x)}{\mathrm{d}\mu_x^f(y)} \right) \mathrm{d}\pi^*(y|x)\mathrm{d}\mathbb{P}(x) =$$

$$\int_{\mathcal{X}} \int_{\mathcal{Y}} \log \left( \frac{\mathrm{d}\pi^*(y|x)\mathrm{d}\mathbb{P}(x)}{\mathrm{d}\mu_x^f(y)\mathrm{d}\mathbb{P}(x)} \right) \mathrm{d}\pi^*(y|x)\mathrm{d}\mathbb{P}(x) =$$

$$\int_{\mathcal{X} \times \mathcal{Y}} \log \left( \frac{\mathrm{d}\pi^*(x,y)}{\mathrm{d}\pi^f(x,y)} \right) \mathrm{d}\pi^*(x,y) = \mathrm{KL} \left( \pi^* \| \pi^f \right) ,$$

which completes the proof. $\qquad \square$

## B.3 PROOF OF THEOREM 3

*Proof.* The direct derivations for (18) read as follows:

$$\frac{\partial}{\partial\theta} L(\theta) = -\varepsilon \int_{\mathcal{X}} \frac{\partial}{\partial\theta} \log Z(f_\theta, x)\mathrm{d}\mathbb{P}(x) + \int_{\mathcal{Y}} \frac{\partial}{\partial\theta} f_\theta(y)\mathrm{d}\mathbb{Q}(y) =$$

$$-\varepsilon \int_{\mathcal{X}} \frac{1}{Z(f_\theta, x)} \left\{ \frac{\partial}{\partial\theta} \int_{\mathcal{Y}} \exp \left( \frac{f_\theta(y) - c(x,y)}{\varepsilon} \right) \mathrm{d}y \right\} \mathrm{d}\mathbb{P}(x) + \int_{\mathcal{Y}} \frac{\partial}{\partial\theta} f_\theta(y)\mathrm{d}\mathbb{Q}(y) =$$

$$-\varepsilon \int_{\mathcal{X}} \left\{ \frac{1}{Z(f_\theta, x)} \int_{\mathcal{Y}} \left[ \frac{\frac{\partial}{\partial\theta} f_\theta(y)}{\varepsilon} \right] \exp \left( \frac{f_\theta(y) - c(x,y)}{\varepsilon} \right) \mathrm{d}y \right\} \mathrm{d}\mathbb{P}(x) + \int_{\mathcal{Y}} \frac{\partial}{\partial\theta} f_\theta(y)\mathrm{d}\mathbb{Q}(y) =$$

$$-\int_{\mathcal{X}} \left\{ \int_{\mathcal{Y}} \left[ \frac{\partial}{\partial\theta} f_\theta(y) \right] \underbrace{\frac{1}{Z(f_\theta, x)} \exp \left( \frac{f_\theta(y) - c(x,y)}{\varepsilon} \right) \mathrm{d}y}_{=\mathrm{d}\mu_x^{f_\theta}(y)} \right\} \mathrm{d}\mathbb{P}(x) + \int_{\mathcal{Y}} \frac{\partial}{\partial\theta} f_\theta(y)\mathrm{d}\mathbb{Q}(y) =$$

$$-\int_{\mathcal{X}} \int_{\mathcal{Y}} \left[ \frac{\partial}{\partial\theta} f_\theta(y) \right] \mathrm{d}\mu_x^{f_\theta}(y)\mathrm{d}\mathbb{P}(x) + \int_{\mathcal{Y}} \frac{\partial}{\partial\theta} f_\theta(y)\mathrm{d}\mathbb{Q}(y) ,$$

which finishes the proof. $\qquad \square$

## B.4 PROOF OF THEOREM 4

To begin with, for the ease of reading and comprehension, we recall the statistical learning setup from §3.3. In short, $X_N$ and $Y_M$ are empirical samples from $\mathbb{P}$ and $\mathbb{Q}$, respectively, $\mathcal{F} \subset \mathcal{C}(\mathcal{Y})$ is a function class in which we are looking for a potential $\widehat{f}$, which optimizes *the empirical weak dual objective*

$$\widehat{F}_{C_{\mathrm{EOT}}}^w(f) = -\varepsilon \frac{1}{N} \sum_{n=1}^{N} \log Z(f, x_n) + \frac{1}{M} \sum_{m=1}^{M} f(y_m),$$

i.e., $\widehat{f} = \arg\max_{f \in \mathcal{F}} \widehat{F}^w_{C_{\text{EOT}}}(f)$. Additionally, we introduce $f^{\mathcal{F}} \overset{\text{def}}{=} \arg\max_{f \in \mathcal{F}} F^w_{C_{\text{EOT}}}(f)$. It optimizes weak dual objective (9) in the function class under consideration. Following statistical generalization practices, our analysis utilizes Rademacher complexity. We recall the definition below.

**Definition 1** (Rademacher complexity $\mathcal{R}_N(\mathcal{F}, \mu)$). *Let $\mathcal{F}$ be a function class and $\mu$ be a distribution. The Rademacher complexity of $\mathcal{F}$ with respect to $\mu$ of sample size $N$ is*

$$\mathcal{R}_N(\mathcal{F}, \mu) \overset{\text{def}}{=} \frac{1}{N} \mathbb{E}\left\{ \sup_{f \in \mathcal{F}} \sum_{n=1}^N f(x_n)\sigma_n \right\},$$

*where $\{x_n\}_{n=1}^N \sim \mu$ are mutually independent, $\{\sigma_n\}_{n=1}^N$ are mutually independent Rademacher random variables, i.e., $Prob(\{\sigma_n = 1\}) = Prob(\{\sigma_n = -1\}) = 0.5$, and the expectation is taken with respect to both $\{x_n\}_{n=1}^N$ and $\{\sigma_n\}_{n=1}^N$.*

Now we are ready to verify our Theorem 4.

*Proof.* Our Theorem 2 yields that

$$\varepsilon \text{KL}\left(\pi^* \| \pi^{\widehat{f}}\right) = F^{w,*}_{C_{\text{EOT}}} - F^w_{C_{\text{EOT}}}(\widehat{f}).$$

Below we upper-bound the right-hand side of the equation above.

$$F^{w,*}_{C_{\text{EOT}}} - F^w_{C_{\text{EOT}}}(\widehat{f}) =$$
$$F^{w,*}_{C_{\text{EOT}}} - F^w_{C_{\text{EOT}}}(f^{\mathcal{F}}) + F^w_{C_{\text{EOT}}}(f^{\mathcal{F}}) - \widehat{F}^w_{C_{\text{EOT}}}(\widehat{f}) + \widehat{F}^w_{C_{\text{EOT}}}(\widehat{f}) - F^w_{C_{\text{EOT}}}(\widehat{f}) \leq$$
$$\left| F^{w,*}_{C_{\text{EOT}}} - F^w_{C_{\text{EOT}}}(f^{\mathcal{F}}) \right| + \tag{31}$$
$$\left| F^w_{C_{\text{EOT}}}(f^{\mathcal{F}}) - \widehat{F}^w_{C_{\text{EOT}}}(\widehat{f}) \right| + \tag{32}$$
$$\left| \widehat{F}^w_{C_{\text{EOT}}}(\widehat{f}) - F^w_{C_{\text{EOT}}}(\widehat{f}) \right| . \tag{33}$$

**Analysis of** (31). Equation (31) relates to *approximation error* and depends on the richness of class $\mathcal{F}$. The detailed investigation of how could the approximation error be treated in the context of our Energy-guided EOT setup and, more generally, EBMs is an interesting avenue for future work.

**Analysis of** (32). Similar to (Taghvaei & Jalali, 2019, Theorem 3.4), we estimate (32) using the Rademacher complexity bounds. First, we need the following technical Lemma.

**Lemma 1.** *For each particular samples $X_N, Y_M$, there exists $\tilde{f} \in \mathcal{F}$, such that:*

$$\left| F^w_{C_{EOT}}(f^{\mathcal{F}}) - \widehat{F}^w_{C_{EOT}}(\widehat{f}) \right| \leq \left| F^w_{C_{EOT}}(\tilde{f}) - \widehat{F}^w_{C_{EOT}}(\tilde{f}) \right|$$

*Proof.* Let's consider $\left| F^w_{C_{\text{EOT}}}(f^{\mathcal{F}}) - \widehat{F}^w_{C_{\text{EOT}}}(\widehat{f}) \right|$. There are two possibilities.

1. $F^w_{C_{\text{EOT}}}(f^{\mathcal{F}}) \geq \widehat{F}^w_{C_{\text{EOT}}}(\widehat{f})$.
   Since $\forall f \in \mathcal{F} : \widehat{F}^w_{C_{\text{EOT}}}(\widehat{f}) \geq \widehat{F}^w_{C_{\text{EOT}}}(f)$, and, in particular, $\widehat{F}^w_{C_{\text{EOT}}}(\widehat{f}) \geq \widehat{F}^w_{C_{\text{EOT}}}(f^{\mathcal{F}})$, it holds:

   $$\left| F^w_{C_{\text{EOT}}}(f^{\mathcal{F}}) - \widehat{F}^w_{C_{\text{EOT}}}(\widehat{f}) \right| = F^w_{C_{\text{EOT}}}(f^{\mathcal{F}}) - \widehat{F}^w_{C_{\text{EOT}}}(\widehat{f}) \leq$$
   $$F^w_{C_{\text{EOT}}}(f^{\mathcal{F}}) - \widehat{F}^w_{C_{\text{EOT}}}(f^{\mathcal{F}}) \leq \left| F^w_{C_{\text{EOT}}}(f^{\mathcal{F}}) - \widehat{F}^w_{C_{\text{EOT}}}(f^{\mathcal{F}}) \right|,$$

   i.e., we set $\tilde{f} \leftarrow f^{\mathcal{F}}$.

2. $\widehat{F}^w_{C_{\text{EOT}}}(\widehat{f}) > F^w_{C_{\text{EOT}}}(f^{\mathcal{F}})$.
   Similar to the previous case, $\forall f \in \mathcal{F} : F^w_{C_{\text{EOT}}}(f^{\mathcal{F}}) \geq F^w_{C_{\text{EOT}}}(f) \Rightarrow F^w_{C_{\text{EOT}}}(f^{\mathcal{F}}) \geq F^w_{C_{\text{EOT}}}(\widehat{f})$. Analogous derivations read as:

   $$\left| F^w_{C_{\text{EOT}}}(f^{\mathcal{F}}) - \widehat{F}^w_{C_{\text{EOT}}}(\widehat{f}) \right| = \widehat{F}^w_{C_{\text{EOT}}}(\widehat{f}) - F^w_{C_{\text{EOT}}}(f^{\mathcal{F}}) \leq$$
   $$\widehat{F}^w_{C_{\text{EOT}}}(\widehat{f}) - F^w_{C_{\text{EOT}}}(\widehat{f}) \leq \left| F^w_{C_{\text{EOT}}}(\widehat{f}) - \widehat{F}^w_{C_{\text{EOT}}}(\widehat{f}) \right|,$$

   i.e., we set $\tilde{f} \leftarrow \widehat{f}$ and finish the proof of the Lemma.

$$\square$$

Now we continue the proof of our Theorem. For particular samples $X_N$ and $Y_M$, consider $\tilde{f}$ from Lemma 1. We derive:

$$\left| F_{C_{\text{EOT}}}^w(f^{\mathcal{F}}) - \widehat{F}_{C_{\text{EOT}}}^w(\widehat{f}) \right| \leq \left| F_{C_{\text{EOT}}}^w(\tilde{f}) - \widehat{F}_{C_{\text{EOT}}}^w(\tilde{f}) \right| =$$

$$\left| \int_{\mathcal{X}} \left[ -\varepsilon \log Z(\tilde{f}, x) \right] \mathrm{d}\mathbb{P}(x) + \int_{\mathcal{Y}} \tilde{f}(y) \mathrm{d}\mathbb{Q}(y) - \left\{ \sum_{n=1}^{N} \frac{-\varepsilon \log Z(\tilde{f}, x_n)}{N} + \sum_{m=1}^{M} \frac{\tilde{f}(y_m)}{M} \right\} \right| =$$

$$\left| \left\{ \int_{\mathcal{X}} \left[ -\varepsilon \log Z(\tilde{f}, x) \right] \mathrm{d}\mathbb{P}(x) - \sum_{n=1}^{N} \frac{-\varepsilon \log Z(\tilde{f}, x_n)}{N} \right\} + \left\{ \int_{\mathcal{Y}} \tilde{f}(y) \mathrm{d}\mathbb{Q}(y) - \sum_{m=1}^{M} \frac{\tilde{f}(y_m)}{M} \right\} \right| \leq$$

$$\left| \int_{\mathcal{X}} \left[ -\varepsilon \log Z(\tilde{f}, x) \right] \mathrm{d}\mathbb{P}(x) - \sum_{n=1}^{N} \frac{-\varepsilon \log Z(\tilde{f}, x_n)}{N} \right| + \left| \int_{\mathcal{Y}} \tilde{f}(y) \mathrm{d}\mathbb{Q}(y) - \sum_{m=1}^{M} \frac{\tilde{f}(y_m)}{M} \right| \leq$$

$$\sup_{f \in \mathcal{F}} \left| \int_{\mathcal{X}} f^{C_{\text{EOT}}}(x) \mathrm{d}\mathbb{P}(x) - \sum_{n=1}^{N} \frac{f^{C_{\text{EOT}}}(x_n)}{N} \right| + \sup_{f \in \mathcal{F}} \left| \int_{\mathcal{Y}} f(y) \mathrm{d}\mathbb{Q}(y) - \sum_{m=1}^{M} \frac{f(y_m)}{M} \right| =$$

$$\sup_{h \in \mathcal{F}^{C_{\text{EOT}}}} \left| \int_{\mathcal{X}} h(x) \mathrm{d}\mathbb{P}(x) - \sum_{n=1}^{N} \frac{h(x_n)}{N} \right| + \sup_{f \in \mathcal{F}} \left| \int_{\mathcal{Y}} f(y) \mathrm{d}\mathbb{Q}(y) - \sum_{m=1}^{M} \frac{f(y_m)}{M} \right|.$$

Recall that $\mathcal{F}^{C_{\text{EOT}}} = \{f^{C_{\text{EOT}}} : f \in \mathcal{F}\} = \{-\varepsilon \log Z(f, \cdot) : f \in \mathcal{F}\}$. Thanks to well-known Rademacher bound (Shalev-Shwartz & Ben-David, 2014, Lemma 26.2), it holds:

$$\mathbb{E} \left\{ \sup_{h \in \mathcal{F}^{C_{\text{EOT}}}} \left| \int_{\mathcal{X}} h(x) \mathrm{d}\mathbb{P}(x) - \sum_{n=1}^{N} \frac{h(x_n)}{N} \right| \right\} \leq 2 \mathcal{R}_N(\mathcal{F}^{C_{\text{EOT}}}, \mathbb{P}),$$

$$\mathbb{E} \left\{ \sup_{f \in \mathcal{F}} \left| \int_{\mathcal{Y}} f(y) \mathrm{d}\mathbb{Q}(y) - \sum_{m=1}^{M} \frac{f(y_m)}{M} \right| \right\} \leq 2 \mathcal{R}_M(\mathcal{F}, \mathbb{Q}),$$

where the expectations are taken with respect to samples $X_N$ and $Y_M$. Combining the results above, we conclude:

$$\mathbb{E} \left| F_{C_{\text{EOT}}}^w(f^{\mathcal{F}}) - \widehat{F}_{C_{\text{EOT}}}^w(\widehat{f}) \right| \leq 2 \mathcal{R}_N(\mathcal{F}^{C_{\text{EOT}}}, \mathbb{P}) + 2 \mathcal{R}_M(\mathcal{F}, \mathbb{Q}). \tag{34}$$

**Analysis of** (33). Similar to the previous case, we obtain the inequality:

$$\left| \widehat{F}_{C_{\text{EOT}}}^w(\widehat{f}) - F_{C_{\text{EOT}}}^w(\widehat{f}) \right| \leq$$

$$\sup_{h \in \mathcal{F}^{C_{\text{EOT}}}} \left| \int_{\mathcal{X}} h(x) \mathrm{d}\mathbb{P}(x) - \sum_{n=1}^{N} \frac{h(x_n)}{N} \right| + \sup_{f \in \mathcal{F}} \left| \int_{\mathcal{Y}} f(y) \mathrm{d}\mathbb{Q}(y) - \sum_{m=1}^{M} \frac{f(y_m)}{M} \right|.$$

Therefore,

$$\mathbb{E} \left| \widehat{F}_{C_{\text{EOT}}}^w(\widehat{f}) - F_{C_{\text{EOT}}}^w(\widehat{f}) \right| \leq 2 \mathcal{R}_N(\mathcal{F}^{C_{\text{EOT}}}, \mathbb{P}) + 2 \mathcal{R}_M(\mathcal{F}, \mathbb{Q}). \tag{35}$$

By gathering equations (31, 34, 35), we prove the theorem:

$$\mathbb{E} \left[ \text{KL} \left( \pi^* \| \pi^{\widehat{f}} \right) \right] = \varepsilon^{-1} \mathbb{E} \left\{ F_{C_{\text{EOT}}}^{w,*} - F_{C_{\text{EOT}}}^w(\widehat{f}) \right\} \leq$$

$$\varepsilon^{-1} \left| F_{C_{\text{EOT}}}^{w,*} - F_{C_{\text{EOT}}}^w(f^{\mathcal{F}}) \right| + \varepsilon^{-1} \mathbb{E} \left| F_{C_{\text{EOT}}}^w(f^{\mathcal{F}}) - \widehat{F}_{C_{\text{EOT}}}^w(\widehat{f}) \right| + \varepsilon^{-1} \mathbb{E} \left| \widehat{F}_{C_{\text{EOT}}}^w(\widehat{f}) - F_{C_{\text{EOT}}}^w(\widehat{f}) \right| \leq$$

$$\varepsilon^{-1} \left| F_{C_{\text{EOT}}}^{w,*} - F_{C_{\text{EOT}}}^w(f^{\mathcal{F}}) \right| + \varepsilon^{-1} \left[ 4 \mathcal{R}_N(\mathcal{F}^{C_{\text{EOT}}}, \mathbb{P}) + 4 \mathcal{R}_M(\mathcal{F}, \mathbb{Q}) \right].$$

$$\square$$

# C  EXTENDED EXPERIMENTS

## C.1  COLORED MNIST

In this subsection, we consider Colored MNIST (Gushchin et al., 2023, §5.3). Following (Gushchin et al., 2023), we set source and target distributions $\mathbb{P}$ and $\mathbb{Q}$ to be colored handwritten images of digits "2" and "3" accordingly. The entropic regularization coefficients are in range $\varepsilon = 0.01, 0.1, 1$.

The qualitative results of learning our model directly in the data space are presented in Figure 4. As we can see, our learned EOT plans successfully preserve color and geometry of the transformed images. Generated images (Figures 4b, 4c, 4d) are slightly noised since we add noise to target images when training for

| $\epsilon$ | 0.01 | 0.1 | 1 |
|---|---|---|---|
| LPIPS (VGG) | 0.043 | 0.063 | 0.11 |

Table 3: LPIPS variance of **Our** method for ColoredMNIST "2"→"3" transfer.

stability. For quantitative analysis and comparison with competitive methods, we borrow the results on the ColoredMNIST transfer problem for (Bortoli et al., 2021), (Daniels et al., 2021), (Gushchin et al., 2023) from (Gushchin et al., 2023). Additionally, we run the code for the recent (Shi et al., 2023) on our own. The methods generally work for different $\varepsilon$ due to their principles, and we choose $\varepsilon = 1$ as an admissible entropic regularization power for all methods except (Daniels et al., 2021) which struggles for small $\varepsilon$, see the discussion in § 4.2. For it, we choose $\varepsilon = 25$. The obtained FID metrics are reported in Table 4. For the qualitative performance of baselines, see (Gushchin et al., 2023, Fig. 2). Besides, we provide Table 3 with LPIPS metric to show that the diversity of our method increases with $\varepsilon$.

| Method | **Ours** | (Bortoli et al., 2021) | (Shi et al., 2023) | (Gushchin et al., 2023) | (Daniels et al., 2021) |
|---|---|---|---|---|---|
| FID | 109 | 93 | 91.3 | 6.3 | $14.7(\varepsilon = 25)$ |

Table 4: Baselines FID for ColoredMNIST "2"→"3" transfer, $\varepsilon = 1$

We honestly state that the FID of our approach is not good. One reason is that the default Langevin dynamic produces slightly noisy samples. FID is known to terribly react to any noise. Secondly, we emphasize that we adapt the simplest long-run EBMs with persistent replay buffer (Nijkamp et al., 2020) for the experiment, see Appendix D.4 for the details. We leave the applications of modern EBMs which can generate sharp data (Du & Mordatch, 2019; Du et al., 2021) for future research.

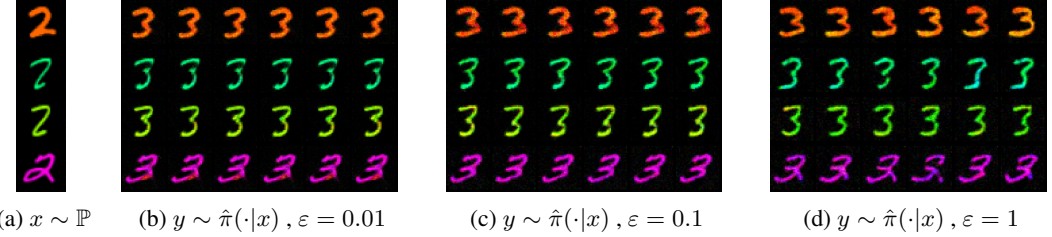

(a) $x \sim \mathbb{P}$  (b) $y \sim \hat{\pi}(\cdot|x)$, $\varepsilon = 0.01$  (c) $y \sim \hat{\pi}(\cdot|x)$, $\varepsilon = 0.1$  (d) $y \sim \hat{\pi}(\cdot|x)$, $\varepsilon = 1$

Figure 4: Quantitative performance of Energy-guided EOT on Colored MNIST.

## C.2 EXTENDED HIGH-DIMENSIONAL UNPAIRED IMAGE-TO-IMAGE TRANSLATION

In this section, we provide additional quantitative results and comparisons for our considered high-dimensional I2I setup. In Table 5, we show **uncurated** samples from our approach learned on $512 \times 512$ AFHQ Cat→Dog and Wild→Dog image transfer problems. To compare our visual results with alternatives, we demonstrate the pictures generated by (Zhao et al., 2022) and (Daniels et al., 2021) solvers, see Figure 6. The former demonstrates SOTA results, see Table 2, but has no relation to OT. The latter is the closest approach to ours. For (Daniels et al., 2021), we trained their algorithm in the same setup as we used, with the latent space of the StylaGAN and transport cost $c(x, y) = \frac{1}{2}\|x - G(z)\|_2^2$, see § 5.3. We found that their method works only for $\varepsilon = 10000$ yielding unconditional generation. It is in concordance with our findings about the approach, see discussion in § 4.2, and the insights from the original work, see (Daniels et al., 2021, §5.1).

## D EXPERIMENTAL DETAILS

**General Details.** For the first two experiments, we take the advantage of replay buffer $\mathcal{B}$ constructed as described in (Du & Mordatch, 2019). When training, the ULA algorithm is initialized by samples from $\mathcal{B}$ with probability $p = 0.95$ and from Gaussian noise with probability $1 - p = 0.05$. For the last two image-data experiments, we also use a similar replay buffer but with $p = 1$, i.e., we do not update $\mathcal{B}$ with short-run samples.

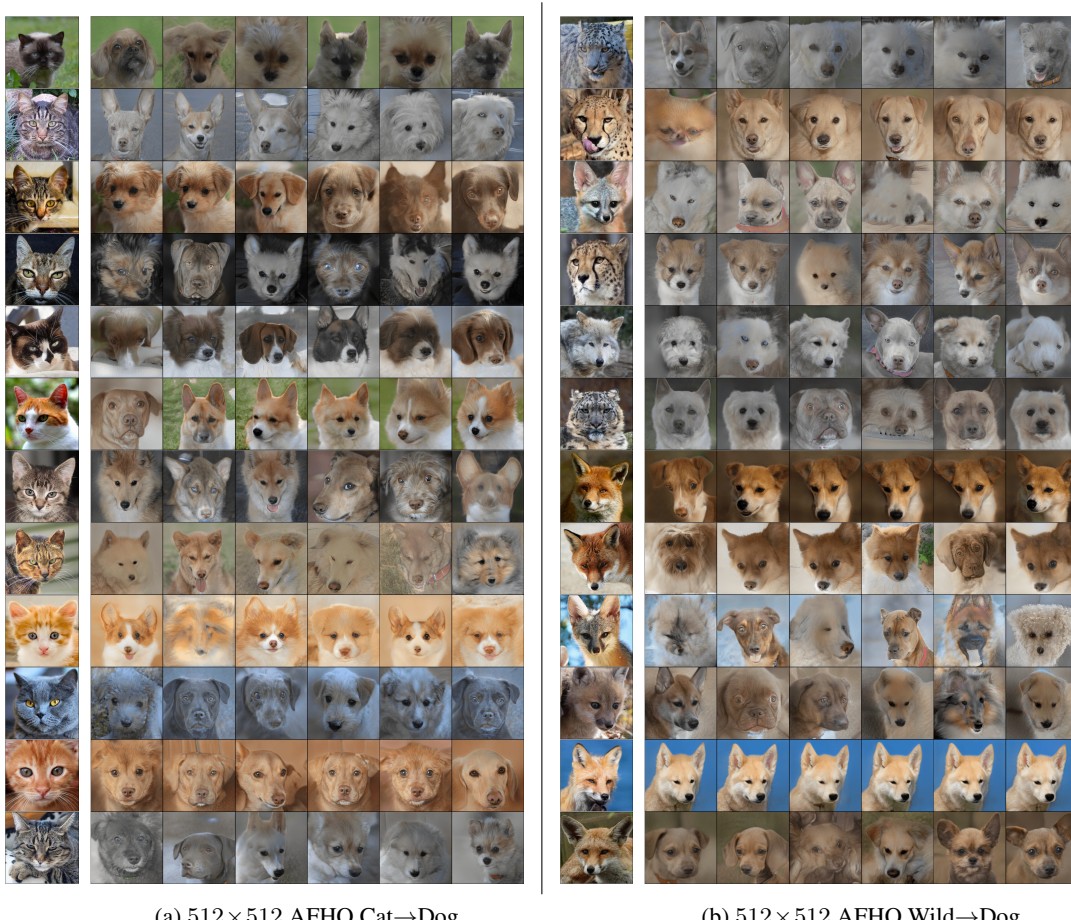

(a) 512×512 AFHQ Cat→Dog      (b) 512×512 AFHQ Wild→Dog

Figure 5: Uncurated Image-to-Image translation by **our** method in the latent space of StyleGAN.

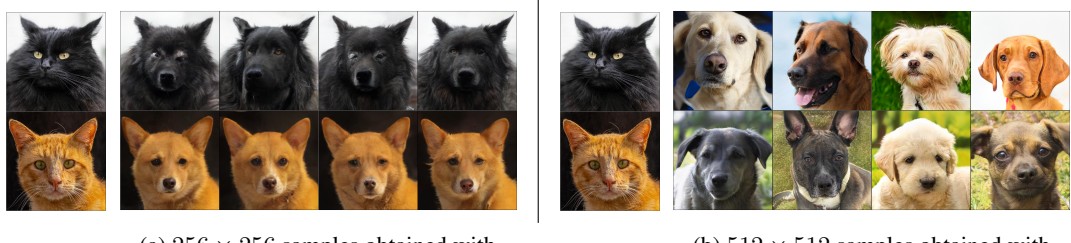

(a) 256 × 256 samples obtained with (Zhao et al., 2022)      (b) 512 × 512 samples obtained with (Daniels et al., 2021), $\varepsilon = 10000$

Figure 6: Image-to-Image Cat→Dog translation by alternative methods

## D.1 TOY 2D DETAILS

We parameterize the potential $f_\theta$ as MLP with two hidden layers and *LeakyReLU*(`negative_slope=` 0.2) as the activation function. Each hidden layer has 256 neurons. The hyperparameters of Algorithm 1 are as follows: $K = 100, \sigma_0 = 1, N = 1024$, see the meaning of each particular variable in the algorithm listing. The Langevin discretization steps are $\eta = 0.05$ for $\varepsilon = 0.1$ and $\eta = 0.005$ for $\varepsilon = 0.001$. The reported numbers are chosen for reasons of the training stability.

**Computation complexity**. The experiment was conducted on a single GTX 1080 Ti and took approximately two hours for each entropy regularization parameter value.

## D.2 GAUSSIAN-TO-GAUSSIAN DETAILS

For the source and target distributions, we choose $\mathbb{P} = \mathcal{N}(0, \Sigma_X)$ and $\mathbb{Q} = \mathcal{N}(0, \Sigma_Y)$ with $\Sigma_X$ and $\Sigma_Y$ chosen at random. For the reproducibility, these parameters are provided in the code.

The details of baseline methods (Table 1) are given in Appendix E. For each particular parameter set $\varepsilon, D$, our trained potential $f_\theta$ is given by MLP with three hidden layers, 512 neurons each, and *ReLU* activation. The same architectural setup is chosen for $\lfloor$Gushchin et. al.$\rceil$ (Gushchin et al., 2023) for fair comparison. The hyperparameters of Algorithm 1 are the same for each $\varepsilon, D$: $K = 100, \sigma_0 = 1, N = 1024, \eta = 0.1$. At the inference stage, we run ULA steps for $K_{\text{test}} = 700$ iterations. The only parameter which we choose to be dependent on $\varepsilon, D$ is the learning rate. It affects the stability of the training. The particular learning rates which are used in our experiments are given in Table 5. More specific training peculiarities could be found in our code.

**Bures-Wasserstein UVP metric**. The $\text{B}\mathcal{W}_2^2$-UVP metric (Korotin et al., 2021b, Eq. 18) is the Wasserstein-2 distance between distributions $\pi_1$ and $\pi_2$ that are coarsened to Gaussians and normalised by the variance of distribution $\pi_2$:

$$\text{B}\mathcal{W}_2^2\text{-UVP}(\pi_1, \pi_2) \stackrel{\text{def}}{=} \frac{100\%}{\frac{1}{2}\text{Var}(\pi_2)} \mathcal{W}_2^2 \big( \mathcal{N}(\mu_{\pi_1}, \Sigma_{\pi_1}), \mathcal{N}(\mu_{\pi_2}, \Sigma_{\pi_2}) \big).$$

In our experiment, $\pi_2$ is the optimal plan $\pi^*$ which is known to be Gaussian, and $\pi_1$ is the learned plan $\hat{\pi}$, whose mean and covariance are estimated by samples.

**Computation complexity**. Each experiment with particular $\varepsilon, D$ takes approximately 12 hours on a single GTX 1080 Ti.

| $D$ | 2 | | | 16 | | | 64 | | | 128 | | |
|---|---|---|---|---|---|---|---|---|---|---|---|---|
| $\varepsilon$ | 0.1 | 1 | 10 | 0.1 | 1 | 10 | 0.1 | 1 | 10 | 0.1 | 1 | 10 |
| lr | $5 \cdot 10^{-7}$ | $4 \cdot 10^{-7}$ | $2 \cdot 10^{-7}$ | $2 \cdot 10^{-5}$ | $4 \cdot 10^{-6}$ | $1 \cdot 10^{-5}$ | $7 \cdot 10^{-5}$ | $4 \cdot 10^{-5}$ | $2 \cdot 10^{-5}$ | $2 \cdot 10^{-4}$ | $5 \cdot 10^{-5}$ | $5 \cdot 10^{-5}$ |

Table 5: Learning rates for Gaussian-to-Gaussian experiment; $\varepsilon = 0.1, 1, 10$ and $D = 2, 16, 64, 128$.

## D.3 HIGH-DIMENSIONAL UNPAIRED IMAGE-TO-IMAGE TRANSLATION DETAILS

**General details.** In this experiment, we learn EOT between a source distribution of images $\mathbb{P} \in \mathcal{P}(\mathbb{R}^{3 \times 512 \times 512})$ and $\mathbb{Q} = \mathcal{N}(0, I_{512}) \in \mathcal{P}(\mathbb{R}^{512})$, which is the latent distribution of the pretrained StyleGAN $G$. We use non-euclidean cost $c(x, y) = \frac{1}{2} \|x - G(z)\|_2^2$. Below we describe the primary idea behind this choice. Consider the pushforward distribution $\mathbb{Q}^{\text{ambi}} \stackrel{\text{def}}{=} G_\sharp \mathbb{Q}$. In our case, it is the parametric distribution of AFHQ Dogs. Thanks to our specific cost function, the learned optimal conditional plans $\pi^{\hat{f}}(\cdot|x)$ between $\mathbb{P}$ and $\mathbb{Q}$ help to approximate (given $\varepsilon$ is sufficiently small) the *standard Euclidean* Optimal Transport between $\mathbb{P}$ and $\mathbb{Q}^{\text{ambi}}$, which is the motivating problem of several OT researches (Makkuva et al., 2020; Korotin et al., 2021a). The corresponding (stochastic) mapping is given by pushforward distributions $G_\sharp \pi^{\hat{f}}(\cdot|x)$. In practice, we sample from $\pi^{\hat{f}}(\cdot|x)$ using our cost-guided MCMC and then pass the obtained samples through $G$. Note that our setup seems to be the first theoretically-advised attempt to leverage $\mathbb{W}_2^2$ OT between $512 \times 512$ images.

**Technical details**. The AFHQ dataset is taken from the StarGAN v2 (Choi et al., 2020) github:

```
https://github.com/clovaai/stargan-v2.
```

The dataset includes three groups of high-quality $512 \times 512$ images: Dogs, Cats and Wilds (wildlife animals). The latter two groups are used as the source distributions $\mathbb{P}$. The pretrained (on AFHQ Dogs) StyleGAN2-ADA Karras et al. (2020) is taken from the official PyTorch implementation:

```
https://github.com/NVlabs/stylegan2-ada-pytorch.
```

As a potential $f_\theta$ which operates in the 512-dimensional latent space of the StyleGAN model, we choose fully-connected MLP with *ReLU* activations and three hidden layers with $1024, 512$ and $256$ neurons, accordingly. The training hyperparameters are: $K = 100, \sqrt{\eta} = 0.008, \sigma_0 = 1.0, N = 128$. For both Cat$\rightarrow$Dog and Wild$\rightarrow$Dog experiments, we train our model for 11 epochs with a learning

rate $10^{-4}$ which starts monotonically decreasing to $10^{-5}$ after the fifth epoch. At inference, we initialize the sampling procedure (in the latent space) with standard Normal noise. Then we repeat Langevin steps for $K_{\text{test}}^{\text{init}} = 1000$ iterations with $\sqrt{\eta} = 0.008$. After that, additional $K_{\text{test}}^{\text{refine}} = 1000$ steps are performed with $\sqrt{\eta} = 0.0005$. The obtained latent codes are then passed through the StyleGAN generator, yielding the images from the AFHQ Dog dataset.

**Computation complexity**. The training takes approximately a day on 4 A100 GPUs. The inference (as described above) takes about 20 minutes per batch on a single A100 GPU.

### D.4 COLORED MNIST DETAILS

For generating Colored MNIST dataset, we make use of the code, provided by the authors of (Gushchin et al., 2023). The dataset consists of colored images of digit "2" ($\approx$ 7K) and colored images of digit "3" ($\approx$ 7K). The images are scaled to resolution $32 \times 32$.

To solve the problem in view, we adapt the base EBM code from (Nijkamp et al., 2020):

$$\texttt{https://github.com/point0bar1/ebm-anatomy.}$$

We do nothing but embed the cost function gradient's computation when performing Langevin steps, leaving all the remaining technical details unchanged. In particular, we utilize simple CNNs with *LeakyReLU*(`negative_slope`= 0.05) activations as our learned potential $f_\theta$. We pick batch size $N = 256$ and initialize the persistent replay buffer at random using *Uniform*$[-1, 1]^{3 \times 32 \times 32}$ distribution. We use *Adam* optimizer with learning rate gradually decreasing from $3 \cdot 10^{-5}$ to $10^{-5}$. The reported images 4 correspond to approximately 7000 training iterations.

For each entropic coefficient $\varepsilon = 0.01, 0.1, 1$, we run 6 experiments with the parameters given in Table 6. For training stability, we add Gaussian noise $\mathcal{N}(0, 9 \cdot \eta)$ to target samples when computing loss estimate $\hat{L}$ in Algorithm 1.

| $\varepsilon$ | 0.01 | 0.1 | 1 |
|---|---|---|---|
| $K \in$ | $\{500, 1000, 2000\}$ | $\{500, 1000, 2000\}$ | $\{500, 1000, 2000\}$ |
| $\sqrt{\eta} \in$ | $\{0.1, 0.3\}$ | $\{0.1, 0.3\}$ | $\{0.2, 0.3\}$ |

Table 6: Training parameters for Colored MNIST; $\varepsilon = 0.01, 0.1, 1$.

At the inference stage, we initialize the MCMC chains with source data samples. The ULA steps are repeated for $K_{\text{test}} = 2000$ iterations with the same Langevin discretization step size $\eta$ as the one used at the training stage. For each $\varepsilon$, the reported images 4 are picked for those parameters set $K, \eta$, which we found to be the best in terms of qualitative performance.

For LPIPS calculation, we use the official code:

$$\texttt{https://github.com/richzhang/PerceptualSimilarity,}$$

where we pick VGG backbone for calculating lpips features. To calculate the resulting metric, we sample 18 target images from $\pi^{\hat{f}}(\cdot|x)$ for each test source image $x$. For every pair of these 18 images, we compute LPIPS and report the average value (along the generated target images and source ones).

**Computation complexity**. It takes approximately one day on one V100 GPU to complete an image data experiment for each set of parameters.

## E DETAILS OF THE BASELINE METHODS

In this section, we discuss details of the baseline methods with which we compare our method on the Gaussian-to-Gaussian transformation problem.

$\lfloor$**Daniels et.al.**$\rceil$ (Daniels et al., 2021). We use the code from the authors' repository

$$\texttt{https://github.com/mdnls/scones-synthetic}$$

for their evaluation in the Gaussian case. We employ their configuration `blob/main/config.py`.

⌊**Seguy et.al.**⌉ (Seguy et al., 2018). We use the part of the code of SCONES corresponding to learning dual OT potentials `blob/main/cpat.py` and the barycentric projection `blob/main/bproj.py` in the Gaussian case with configuration `blob/main/config.py`.

⌊**Chen et.al.**⌉ *(Joint)* (Chen et al., 2022). We utilize the official code from

$$\text{https://github.com/ghliu/SB-FBSDE}$$

with their configuration `blob/main/configs/default_checkerboard_config.py` for the checkerboard-to-noise toy experiment, changing the number of steps of dynamics from 100 to 200 steps. Since their hyper-parameters are developed for their 2-dimensional experiments, we increase the number of iterations for dimensions 16, 64 and 128 to 15 000.

⌊**Chen et.al.**⌉ *(Alt)* (Chen et al., 2022). We also take the code from the same repository as above. We base our configuration on the authors' one (`blob/main/configs/default_moon_to_spiral_config.py`) for the moon-to-spiral experiment. As earlier, we increase the number of steps of dynamics up to 200. Also, we change the number of training epochs for dimensions 16, 64 and 128 to 2,4 and 8 correspondingly.

⌊**De Bortoli et.al.**⌉ (Bortoli et al., 2021). We utilize the official code from

$$\text{https://github.com/JTT94/diffusion\_schrodinger\_bridge}$$

with their configuration `blob/main/conf/dataset/2d.yaml` for toy problems. We increase the amount of steps of dynamics to 200 and the number of steps of IPF procedure for dimensions 16, 64 and 128 to 30, 40 and 60, respectively.

⌊**Vargas et.al.**⌉ (Vargas et al., 2021). We use the official code from

$$\text{https://github.com/franciscovargas/GP\_Sinkhorn}$$

with hyper-parameters from `blob/main/notebooks/2D Toy Data/2d_examples.ipynb`. We set the number of steps to 200. As earlier, we increase the number of steps of IPF procedure for dimensions 16, 64 and 128 to 1000, 3500 and 5000, respectively.

⌊**Vargas et.al.**⌉ (Vargas et al., 2021). We tested the official code from

$$\text{https://github.com/franciscovargas/GP\_Sinkhorn}$$

Instead of Gaussian processes, we used a neural network as for ⌊**ENOT**⌉. We use $N = 200$ discretization steps as for other SB solvers, 5000 IPF iterations, and 512 samples from distributions $\mathbb{P}_0$ and $\mathbb{P}_1$ in each of them. We use the Adam optimizer with $lr = 10^{-4}$ for optimization.

⌊**Gushchin et.al.**⌉ (Gushchin et al., 2023) We use the code provided by the authors. In our experiments, we use exactly the same hyperparameters for this setup as the authors (Gushchin et al., 2023, Appendix B), except the number of discretization steps $N$, which we set to 200 as well as for other Schrödinger Bridge based methods.

## F EXTENDED DISCUSSION OF LIMITATIONS

In general, the main limitation of our approach is the usage of MCMC. This procedure is time-consuming and requires adjusting several hyperparameters. Moreover, in practice, it may not always converge to the desired distribution which introduces additional biases. The other details of launching our proposed algorithm arise due to its connection to EBM's learning procedure. It is known that EBMs for generative modelling could be trained by two different optimization regimes: short-run (non-convergent) training and long-run (convergent) training (Nijkamp et al., 2020). In the first regime, the learned potential does not necessarily represent the energy function of the learned distribution. Because of this, the short-run mode may not always be adapted for Energy-guided EOT, since it seems crucial for $f_\theta$ to represent the true component of the conditional Energy potentials $E_{\mu_x^{f_\theta}}(y) = \frac{c(x,y)-f_\theta(y)}{\varepsilon}$. In particular, for our *Colored MNIST* experiment, we found the short-run regime to be unstable and utilize exclusively long-run mode. At the same time, for

moderate-dimensional *Toy 2D* and *Gaussian-to-Gaussian* experiments as well as for latent-space *high-dimensional I2I setup*, non-convergent training was successful.

