# OpenReview forum: "Energy-guided Entropic Neural Optimal Transport"
_ICLR.cc/2024/Conference — ICLR 2024 poster_

### Official Review · Reviewer_2Kak · 2023-10-29

**Soundness:** 3 good
**Presentation:** 2 fair
**Contribution:** 3 good
**Rating:** 6
**Confidence:** 4

**Summary:**

This work bridges the gap between Energy-Based Models (EBMs) and Entropy-regularized Optimal Transport (EOT). In particular, it demonstrates that solving EOT is, to some extent, equivalent to EBM training. Thus, a novel methodology is introduced that leverages recent EBM developments to enhance the EOT solver. The approach is theoretically underpinned by generalization bounds and validated through practical applications in 2D and image domains.

**Strengths:**

This research uncovers the connection between energy-based models and entropy-regularized optimal transport, opening up new applications for EBMs, including unpaired data-to-data translation.

**Weaknesses:**

My primary concern centres around the scalability of the proposed approach. The training process hinges on simulating MCMC, which poses significant challenges when dealing with high-dimensional datasets. While promising results have been demonstrated in experiments on high-dimensional unpaired image-to-image translation, it is worth noting that this approach couples with a pretrained GAN model and conducts training in latent spaces. I hold reservations about its direct applicability to image spaces.

**Questions:**

- It appears that equation 18 is akin to the maximum likelihood training of the EBM defined in equation 13. Could you offer a more intuitive explanation of why maximizing the likelihood of equation 13 is valid and how it equates to entropy-regularized optimal transport?
- Can equation 17 be optimised using alternative EBM training techniques, such as noise contrastive estimate, and score matching?

---

> ### Author Response · Authors · 2023-11-19
> **Answers to Reviewer 2Kak. Part 1**
>
> Dear reviewer! Thank you for your positive feedback and profound questions. Below we answer your questions and comments.
>
> **(1) Scalability of the proposed approach. Direct applicability of the approach to image spaces**
>
> To begin with, we want to highlight that there exist modern EBM models which are capable of generating high-fidelity image data, see, e.g., [Zhao], [Du]. This fact serves as an indirect witness that our method is scalable to such complex data setups as well. We assume that the combination of the aforementioned advanced techniques with our proposed methodology may yield Energy-based EOT solvers which operates in high-dimensional image spaces directly without StyleGAN proxy. However, in our paper we stick more to the theory and methodology rather than technical stuff with hyperparameters/regularization methods/technical tricks selection and leave these adaptations to a future work.
>
> At the same time, we agree with the reviewer that it is really important to showcase the applicability for the method on the image data. That is why we conduct an experiment on ColoredMNIST "2"$\rightarrow$"3" $3 \times 32 \times 32$ digits transfer and demonstrate the results in Appendix C.1. The results are not so good. However, please note that for this experiment we adapt the code from [Nijkamp] - it is probably **the simplest EBM setup** for learning image data. In particular, we parameterize the learned potential $f_{\theta}$ as the trivial Conv2d -> LeakyReLU -> Conv2d -> LeakyReLU -> $\dots$ combination. See our code and Appendices C.1 and D.4 for the details. We humbly think that this experiment already demonstrates the applicability of our methodology, as "it is", without engineering improvements, to image data.
>
> [Nijkamp] Nijkamp et. al., On the Anatomy of MCMC-Based Maximum Likelihood Learning of Energy-Based Models.
>
> **(1.1) "[...]it is worth noting that this approach couples with a pretrained GAN model and conducts training in latent spaces.**
>
> We want to leave a comment regarding our unpaired I2I setup in the latent space of StyleGAN. We want to underline that our Energy-based approach is not a usual method for learning something in a latent space. To the best of our knowledge, a typical method operating in a latent space requires not only a decoder (a model which transforms latent embeddings to data) but also an encoder (which transforms data of interest to the latent embeddings). At the same time, our method *does not need any encoder*.
>
> **(2) Clarifications regarding Eqs. 13 and 18**
>
> We are not sure if we catch the question of the reviewer exactly, but we make a try. At first, we propose to compare objectives (18) and (11).  In fact, the potential $f_{\theta}$ in (18) serves as a test function which compares *the second marginal* of distribution $d \pi^{f_\theta}(x, y) = d \mu_x^{f_\theta}(y) d \mathbb{P}(x)$ and $\mathbb{Q}$. Similarly, the potential $E_{\theta}$ in (11) plays the same role by comparing the generative distribution $\mu_\theta$ and GT distribution $\mu$. In other words, our Energy-based methodology for solving EOT is the combination of **three** principles. We look for a plan $\pi \in \mathcal{P}(\mathcal{X} \times \mathcal{Y})$,
>
> * with the first marginal $\pi(x)$ (projection to $\mathcal{X}$) given by the source distribution $\mathbb{P}$. This condition is satisfied by construction (as we build our plan as $d \pi(x, y) = d \mathbb{P}(x) \cdot d \pi(y \vert x)$).
>  * with the second marginal $\pi(y)$ (projection to $\mathcal{Y}$) given by the target distribution $\mathbb{Q}$. In order to satisfy this condition, we need the optimization given by Eq. (18).
>  * whose conditional distributions $\pi(y \vert x)$ satisfy Eq. (13), i.e., $\pi(y \vert x) = \mu_x^f(y)$ for some potential $f$. This condition is also satisfied by construction. Note, that it is the condition which ensures that plan $\pi$ is EOT plan between its first and second marginals.
>
> Combining principle 3.) with 1.) and 2.), our methodology results in Energy-based EOT solver. Does our comments address your question?

---

> > ### Author Response · Authors · 2023-11-19
> > **Answers to Reviewer 2Kak. Part 2**
> >
> > **(3) Alternative EBM training procedures (noise contrastive estimate, score matching) for EOT**
> >
> > It is the really good and interesting question. We express special gratitude to the reviewer. In short, it seems that these techniques (noise contrastive estimate - NCE and score matching - SM) are not straightforwardly applicable for EOT. The key challenge here is that we aim to model a distribution (in our case, it is the EOT plan $\pi^*$) samples from which **are not known**. This makes our considered setup completely different from a typical generative modeling problem when we **have samples** from an unknown distribution, and we want to somehow approximate the underlining distribution based on these samples. In particular, both NCS and SM require knowledge of samples from the distribution of interest. In fact, in our approach, we manage to alleviate this lack of data by utilizing special properties of EOT plan. It is still an open question whether we can somehow adapt SM-based techniques to solve EOT in a similar way. Well, if you are interested in this topic, we encourage you to think of this question. We are sure that a hypothetical SM-based EOT solver will be of interest to the ML community. Some works which are a good points for inspiration and reflections: [Daniels], [Chen] (the latter utilizes the connection between EOT and the Schrodinger bridge problem).
> >
> > [Zhao] Zhao et. al., Learning Energy-Based Generative Models via Coarse-to-Fine Expanding and Sampling.
> >
> > [Du] Du et. al., Improved Contrastive Divergence Training of Energy-Based Model.
> >
> > [Daniels] Daniels et. al., Score-based generative neural networks for large-scale optimal transport.
> >
> > [Chen] Chen et. al., Stochastic control liaisons: Richard Sinkhorn meets Gaspard Monge on a Schroedinger bridge.
> >
> > **Concluding remarks for the all questions.** We thank the reviewer for the raised concerns and questions. We hope that we properly address them. If so, we kindly ask the reviewer to consider rising the score and confidence. We are ready to continue the discussion of the existing questions, and we are happy to sort out with new one.

---

> > > ### Comment · Reviewer_2Kak · 2023-11-21
> > > **Response to the authors**
> > >
> > > Thanks for your great effort in responding and revising. After reading the rebuttal, I believe that this paper has a good theoretical contribution, which showcases the connection between OT and EBM. This observation is interesting and that is the main reason I am leaning towards accepting the paper. From a practical perspective, the connection to EBM makes the proposed OT solver hard to directly apply in high-dimensional spaces, due to the reliance on MCMC during training. Therefore, compared to other recently developed OT solvers, like Schrodinger bridge, and flow matching, the proposed methods have strong limitations. Overall, I will keep my score unchanged but increase my confidence.

---

### Official Review · Reviewer_7M7Q · 2023-11-06

**Soundness:** 3 good
**Presentation:** 3 good
**Contribution:** 3 good
**Rating:** 6
**Confidence:** 3

**Summary:**

The paper reformulates the weak dual of the energy-optimal transport problem to identify an expectation inside the loss function that results in their loss function being trained in a similar way to energy-based models. This allows energy-based training procedures to be used to solve optimal transport problems.

**Strengths:**

The arguments in the paper are clear and straightforward. The paper is well structured with the contributions highlighted clearly. The background is well-presented and I don't see typos. Figure 2 is well-made and shows the efficacy of their method.

**Weaknesses:**

The biggest weakness of their proposed method uses energy-based training which involves MCMC. I am unclear if this is ideal as MCMC can be tricky. It would be interesting to see if the unpaired image-to-image task can be done with other OT methods to better see how useful this particular formulation and method is.

**Questions:**

Is there a relationship between $(c, \epsilon)$-transform and the Cole–Hopf transformation in PDEs ? They do seem very similar. It would also be very interesting to apply this to more than just continuous distribution but discrete distribution like languages so doing task like language translation would be interesting to see.

---

> ### Author Response · Authors · 2023-11-19
> **Answers to Reviewer 7M7Q. Part 1**
>
> Dear reviewer, thanks for your efforts and your thoughtful review. We appreciate that you have pointed out the methodological and theoretical advantages of our work. Please find answers to your questions and comments below.
>
> **(1) Problems with MCMC training usage**
>
> We agree with the reviewer that our EBMs-based methodology does not provide a "silver bullet" for solving the Entropic Optimal Transport. It indeed may suffer from the difficulties which are common to EBMs. In particular, MCMC procedure can be time-consuming and requires some technical tricks and hyperparameters tuning. Note that we have taken these complications into account in the limitations section. At the same time, EBM is actively developing research direction, and recent works from this area show that EBM can successfully model high-dimensional $512\times 512$ image data in ambient (not latent) space, see, e.g., [Gao], [Zhao]. Adapting advanced EBM techniques from such papers is a fruitful direction for future research.
>
> Second, we would like to draw the reviewer's attention to our high-dimensional I2I translation setup (Section 5.3). In this experiment, we do not experience any difficulties with MCMC stability/convergence, and we use only basic and well-established EBM training techniques, e.g., we use a replay buffer. Nevertheless, we achieve a good quality of learned EOT stochastic mappings. Although this experiment is conducted in the latent space of a pre-trained StyleGAN, we dare to hope that it demonstrates the usefulness of our developed approach as it is, even without substantial technical improvements.
>
> Finally, we would like to emphasise that the main contribution of our work is theoretical and methodological. Here we have managed to establish the link between EBM and EOT and to show that the proposed methodology can be successfully applied in practice. We humbly believe that this is already a good contribution to the ML community.
>
> [Gao] Gao et. al., Learning Energy-Based Models by Diffusion Recovery Likelihood.
>
> [Zhao] Zhao et. al., Learning Energy-Based Generative Models via Coarse-to-Fine Expanding and Sampling.
>
> **(2) Unpaired I2I task with other OT methods**
>
> Our work is not the only approach which can solve Unpaired image-to-image problem following OT formulation. In fact, tackling Unpaired I2I is the primary goal of the vast majority of **continuous** OT-based solvers. We refer the reviewer to Appendix A.2 (subsection "Continuous OT") for the corresponding survey and to Subsection 4.2 which covers the methods specifically designed for continuous EOT. Among them, the work [Korotin] probably demonstrates the most promising results for Unpaired I2I. This work is based on the semi-dual formulation of the OT problem regularised by a negative variance term. It proposes to optimize a max-min **adversarial** objective. This is a drawback of [Korotin], as adversarial learning is tricky and can be unstable. In particular, we couldn't get their algorithm to work properly on the AFHQ Cat$\rightarrow$Dog setup, even downscaled to $64\times 64$ images. Probably, one reason for this is the insufficient amount of data for this problem (5K samples for the source and target domains), since the original paper demonstrates good results on $128 \times 128$ setup of other unpaired I2I problems ($\approx$ 60K samples for the source and target domains). We added the examples of unpaired mapping learned by NOT in the newly updated supplementary materials. In conclusion, we want to underline, that in our paper we consider original AFHQ'$512\times 512$ Cat$\rightarrow$Dog setup, but utilize a pre-trained StyleGAN.
>
> [Korotin] Korotin et. al., Neural Optimal Transport.

---

> > ### Author Response · Authors · 2023-11-19
> > **Answers to Reviewer 7M7Q. Part 2**
> >
> > **(3) Relation between $(c, \epsilon)$-transform and Cole-Hopf transformation**
> >
> > Thank you for this interesting question. At the first glance, the Cole-Hopf transformation does not have a relation to $(c, \epsilon)$-transform because the former is about solving PDEs (partial differential equations) while the latter does not assume any time-varying equations. At the same time, we would like to draw the reviewer's attention to [Caluya]. This work deals with the Schrodinger bridge (SB) problem, which is closely related to EOT. When solving SB, one looks for a specific stochastic process $T$ that starts and ends at pre-defined source and target distributions $\mathbb{P}$ and $\mathbb{Q}$. Under some conditions, if $T^*$ is a solution to the SB problem, then the joint distribution of this stochastic process at the initial and final time points yields the EOT plan between $\mathbb{P}$ and $\mathbb{Q}$. This establishes the connection between EOT and SB. And it turns out that Cole-Hopf transform appears explicitly when solving SB, see [Caluya, Theorem 2]. This theorem states that the potentials defining the Cole-Hopf transformation of the optimal SB process conform to particular PDEs and that the optimal SB drift could be easily derived using these potentials. Probably, the original time-independent $(c, \epsilon)$-transform could be derived as an "marginalized" version of these theoretical observations. However, we are not aware of any works that establish this theoretical link.
> >
> > [Caluya] Caluya et. al., Wasserstein Proximal Algorithms for the Schrodinger Bridge Problem: Density Control with Nonlinear Drift.
> >
> > **(4) Considering discrete (e.g., languages) EOT setup**
> >
> > In general, our methodology could be adapted to discrete setup by considering discrete measures $\mathbb{P}$ , $\mathbb{Q}$ and parameterizing the potential $f$ as a (trainable) vector $f_{\text{discr}} = (f_1, f_2, \dots f_M)$, where $M$ is the number of discrete samples from the target distribution. In this case, the dual loss $L(\theta)$ update (Eq. (18)) could be computed analytically (without MCMC).
> >
> > However, we pose our work as a continuous EOT solver. From our point of view, deriving the discrete version of our methodology and considering discrete applications will spoil the consistency of our work. In fact, there have been developed efficient discrete EOT solvers, see [Peyre\&Cuturi] for the review. Competing with these methods is out of scope of our research.
> >
> > [Peyre\&Cuturi] G. Peyre and M. Cuturi, Computational Optimal Transport.
> >
> > **Concluding remarks for the all questions.** We would be grateful if you could let us know if we have satisfactory answered your questions and addressed your concerns. If so, we kindly ask that you consider increasing your score and confidence. We are also open to discussing any other questions you may have.

---

### Official Review · Reviewer_mVby · 2023-11-08

**Soundness:** 3 good
**Presentation:** 3 good
**Contribution:** 3 good
**Rating:** 6
**Confidence:** 4

**Summary:**

The paper proposes a novel method for computing entropic optimal transport problem (EOT), utilizing techniques from energy-based models (EBM). Specifically, the paper uses the weak dual form to reformulate the EOT problem into an optimization task over space of functions, where the objective takes form similar to the EBM, i.e. in exponential form. The paper then proceeds to parametrize the function space by neural networks, and applies the algorithm to various tasks. Theoretical results are provided regarding why the proposed method approximates the optimal EOT coupling, and how the estimation and approximation contributes to generalization. Experiments are implemented, where various technical subtleties are also addressed.

**Strengths:**

The paper is overall well written and presented, and the the ideas are original to the knowledge of the reviewer. The discussion of all results seem plenty and extensive. Some strengths:
1. The paper points out that the semi dual form of EOT works well for approximation of optimal coupling. The reviewer finds it interesting, as in semi dual form, essentially only one of the two equations of the Schrödinger system is satisfied, thus the other marginal usually lacks control. However, as shown in the theorem 2, even if only one potential is parametrized, constructing a joint distribution by taking conditionals to be normalized exponential models gives clean approximation of the optimal coupling, with bounds on approximation gap. As the construction naturally extrapolates, the proposed method not only computes EOT, but also helps generative sampling.
2. The proposed usage of EBM seems principled, as EOT semi dual form admits exponential form, which enables application of well-studied sampling methods, and the corresponding gradients all have simple feasible forms.
3. The experiments seem plenty and sufficient, and various training techniques for EBM are also discussed.

**Weaknesses:**

Some weaknesses:
1. One major concern is the bound in Theorem 4, where a classical bound of error illustrating the balance between approximation and estimation is provided, using Rademacher complexity and optimality gap. However, it is unclear what is expected as the overall rate from this bound, as the choice of parametric class remains heuristic. This can be important, as generative tasks usually operates in high dimensions, and the dependence in dimensionality seems crucial to justify the applicability. There are plenty of works, for example, please see [1,2] for approximation and estimation using neural networks. Clarifying the proper choice of the parametric class and giving an explicit balancing characterization would give a clearer picture.
2. The paper seems to focus more on applicability of EBM, though this class of methods usually requires sampling, which creates major computational burden and additional estimating error. A full computational and space-time complexity analysis seems needed, as even regardless of the NN optimization, the construction of the loss function requires significant computation to obtain an oracle. Furthermore, it would be interesting to also see how sampling error enters the bound in Theorem 4.

[1] Klusowski, Jason M., and Andrew R. Barron. "Approximation by combinations of ReLU and squared ReLU ridge functions with $\ell^ 1$ and $\ell^ 0$ controls." IEEE Transactions on Information Theory 64.12 (2018): 7649-7656.

[2] Sreekumar, Sreejith, Zhengxin Zhang, and Ziv Goldfeld. "Non-asymptotic performance guarantees for neural estimation of f-divergences." International Conference on Artificial Intelligence and Statistics. PMLR, 2021.

**Questions:**

Please see above (section Weaknesses) for details. An additional question: is it possible to give a characterization of $\epsilon$ dependence of computation/estimation/approximation? The reviewer understands that this is additional work, so simple answers such as exponential/polynomial dependence would be also good.

---

> ### Author Response · Authors · 2023-11-19
> **Answers to Reviewer mVby. Part 1**
>
> Dear reviewer! Thank you for your kind words. We are very pleased that you find our work to be interesting and valuable from both theoretical and practical points of view. Below we comment on your questions.
>
> **(1) Overall rate for the bound from Theorem 4, using explicit rates of NNs approximation, e.g., [1, 2]**
>
> Improving Theorem 4 with explicit numerical bounds is indeed a good question coming from Statistical Learning Theory. However, it seems to be rather non-trivial and deserves a separate research, probably, a separate paper similar to [2]. The key challenge here is the analysis of weak $C_{\text{EOT}}$-transformed functions. This is much more difficult than, say, analysis of functions (Neural Networks) $f$ and their compositions with a pre-definite measurement function $\gamma$ : $\gamma \circ f$. Note that the latter is the setup which is considered in [2]. Therefore, we expect that the derivation of the desired bounds will require substantial theoretical and numerical efforts: definitions, assumptions, theorems, proofs and supporting experiments. It is tantamount to carrying out a distinct research. At the same time, following your suggestions, we cite [1, 2] and indicate the importance of deriving numerical estimation rates for Theorem 4 in the new revision of our manuscript.
>
> We emphasise that our Theorem 4, as given in our paper, is already *important and worth presenting*. It provides a way for estimating the quality of generative model $\pi^{\hat{f}}$ approximating EOT plan from the Statistical Learning Theory perspectives. As we point out in our paper, the majority of works establishing similar statistical bounds deals with other quantities, e.g., the OT (EOT) cost between fixed distributions or the barycentric projections. Note that work [2] also handles discrepancies (several $f$-divergences) between fixed distributions. This is far from the generative modelling setup.
>
> **(2) "The paper seems to focus more on applicability of EBM, [...]." EBM computational and space-time complexity analysis**
>
> Probably, here there is a tiny misunderstanding. Indeed, one of the main aims and contributions of our work is to show that EBMs could be applied (with some modifications) for solving the EOT problem. And we humbly hope that our theoretical findings as well as supporting experiments properly support this thesis. At the same time, we do not set out to answer the question regarding the applicability of EBMs itself, for general-purpose generative modelling problem. We do not answer the question to which extent the existing EBM procedures are efficient or not efficient. In fact, we treat the EBM training procedure as the particular solver for our proposed methodology. We resort to decade-long EBM research. And we are not that heroes who managed to equip EBM with solid theoretical grounds. Note that our Theorem 4 is free of assumptions about the particular, e.g. EBM-advised, method used to optimise the weak dual objective (9).
>
> **(3) Sampling error in Theorem 4**
>
> The sampling error (error due to inexact sampling from EBM due to MCMC) does not enter the bounds in Theorem 4. In fact, Theorem 4 just compares the true EOT plan $\pi^*$ with approximate EOT plan $\pi^{\hat{f}}$ and does not assume any sampling. Note that $\hat{f}$ which appears in Theorem 4 is (by definition) the **perfect** potential which solves weak dual objective (9) given a limited number of empirical samples from the source and target distributions $\mathbb{P}$ and $\mathbb{Q}$ and restricted class of available potentials $\mathcal{F}$. In turn, the sampling error may cause the potential $\hat{f}_{\text{practical}}$ resulting from our practical algorithm to differ from the **perfect** potential $\hat{f}$. Let us call this *optimization error*. Note that MCMC sampling error is not the only source of optimization error. It is also due to stochastic gradient ascent (Eq. 18), usage of batches, etc. The analysis of these quantities is a completely different domain in Machine Learning and out of the scope of our work. As with most generative modelling research, we do not attempt to analyse optimisation errors.

---

> > ### Author Response · Authors · 2023-11-19
> > **Answers to Reviewer mVby. Part 2**
> >
> > **(4) Characterization of $\epsilon$ dependence of computation/estimation/approximation**
> >
> > We thank the reviewer for this interesting question. Our answer is two-fold:
> >
> > **Estimation/approximation.** At the first glance, our Theorem 4 gives clear $\epsilon^{-1}$ dependence for the (averaged) closeness of empirically-recovered EOT plan $\pi^{\hat{f}}$ to the true one. I.e., the smaller the $\epsilon$, the more difficult it gets to recover true EOT plan. It is consistent with the fact that as $\epsilon \rightarrow 0$, the true EOT plan becomes the deterministic OT plan, which can not be recovered using the Energy function. Note, however, that there are some peculiarities which make the picture more complicated.
> >
> > At first, the right-hand side of Eq. (19) involves some quantities, e.g. Rademacher complexity of weak $C_{\text{EOT}}$-transforms, which may implicitly depend on $\epsilon$. The detailed characterization of $\epsilon$ dependence for these quantities is non-trivial and a subject of a follow-up research, as we pointed out when commenting the first raised point.
> >
> > Secondly, one should note that the left-hand side of Eq. (19) also depends on $\epsilon$. We compare a recovered and the true EOT plans exactly for *given* entropic regularization coefficient $\epsilon$. Probably, one conclusion from this remark is that comparing different $\epsilon$ is probably not so important since for different $\epsilon$ we solve different problems and aim at approximating different distributions. Still, the question like "for which $\epsilon$ our methodology may result in a good approximation for the true EOT plan" is of course of high importance.
> >
> > **Computation.** Here we want to add some notes regarding the practical behavior of our training algorithm, first and foremost, Langevin sampling, with regard to $\epsilon$. At first, the conditional Langevin sampling becomes less stable as $\epsilon \rightarrow 0$ because the magnitude of $\nabla \frac{f(y) - c(x, y)}{\epsilon}$ is getting bigger. Secondly, due to limited number of Langevin steps, the resulting Langevin samples better represents less "peaked" conditional distributions $\pi(\cdot \vert x)$ (for larger $\epsilon$) than highly "peaked" ones (for smaller $\epsilon$).
> >
> > In conclusion, both theoretical and practical considerations suggests that the "small" $\epsilon$ regime is less favorable for our methodology than the "large" one. However, deriving the exact behavior is complicated.
> >
> > **Concluding remarks for the all questions**. We thank the reviewer one more time for the interesting and important questions. We did not fully answer them, because some of them are out of scope of our research. We believe that we managed to provide strong arguments to such our opinion. If the reviewer agrees with us, we kindly ask his/her to consider increasing the score and confidence. We are open to follow-up questions.

---

> ### Comment · Reviewer_mVby · 2023-11-20
> **Thanks for the response**
>
> I thank the authors for the response, which addresses all of my questions. The authors are right that I misunderstood a bit -- I was expecting that the training objective uses not samples from Q but the MCMC samples (one comment is that it seems using EBM samples avoids exponentiation in the training, thus adding training stability?). I thus raise confidence and contribution score, and leave the rating unchanged.

---

### Author Response · Authors · 2023-11-19
**General comment to the all reviewers**

Dear reviewers, we want to express our gratitude for dedicating your valuable time to review our paper. Your feedback means a lot to us, and we are delighted that many of you found the theoretical and methodological contributions of our work to be significant and experiments to be sufficient.

The raised questions have entailed only minor changes to the manuscript. In particular:

- We added a comment regarding improving our Theorem 4 with numerical bounds (mVby).

- We empower our supplementary materials with qualitative illustration of the performance of an alternative Unpaired I2I OT solver (7M7Q)

The changes in the **revised** paper are highlighted with blue color. Below you can find the detailed answers.

---

### Meta-Review · Area_Chair_9z9n · 2023-12-06

**Metareview:**

Despite certain concerns, particularly the reliance on MCMC methods during training, which some reviewers noted as a major weakness potentially affecting direct applicability, I recommend the acceptance of this paper. The strengths of the paper, including its well-crafted writing style, the introduction of intriguing ideas such as leveraging the semi-dual form of Entropic Optimal Transport (EOT), and the comprehensive numerical simulations, notably outweigh its limitations. Furthermore, the authors have effectively addressed most of the reviewers' questions in their rebuttal, demonstrating their commitment to refining and clarifying their work. The paper's innovative approach and substantial contribution to its field make it a valuable addition to the academic discourse.

**Justification For Why Not Higher Score:**

In my view the paper is borderline to reject, the contributions are ok but not very strong.

**Justification For Why Not Lower Score:**

In my view the paper is borderline to reject, it could be modified to reject.

---

### Decision · Program_Chairs · 2024-01-16

Accept (poster)